# Global tropospheric effects of aromatic chemistry with the SAPRC-11 mechanism implemented in GEOS-Chem version 9-02

Yingying Yan[1,2], David Cabrera-Perez[3], Jintai Lin[2], Andrea Pozzer[3], Lu Hu[4], Dylan B. Millet[5], William C. Porter[6], Jos Lelieveld[3]

[1] Department of Atmospheric Sciences, School of Environmental Studies, China University of Geosciences (Wuhan), 430074, Wuhan, China

[2] Laboratory for Climate and Ocean-Atmosphere Studies, Department of Atmospheric and Oceanic Sciences, School of Physics, Peking University, Beijing 100871, China

[3] Max-Planck-Institute for Chemistry, Atmospheric Chemistry Department, Mainz, Germany

[4] Department of Chemistry and Biochemistry, University of Montana, Missoula, MT, USA

[5] Department of Soil, Water, and Climate, University of Minnesota, St. Paul, MN, USA

[6] Department of Civil and Environmental Engineering, Massachusetts Institute of Technology, 77 Massachusetts Avenue, Cambridge, MA 02139-4307, USA

Correspondance : Jintai Lin, linjt@pku.edu.cn

## Abstract

The GEOS-Chem model has been updated with the SAPRC-11 aromatics chemical mechanism, with the purpose of evaluating global and regional effects of the most abundant aromatics (benzene, toluene, xylenes) on the chemical species important for tropospheric oxidation capacity. The model evaluation based on surface and aircraft observations indicates good agreement for aromatics and ozone. A comparison between scenarios in GEOS-Chem with simplified aromatic chemistry (as in the standard setup, with no ozone formation from related peroxy radicals or recycling of $NO_x$) and with the SAPRC-11 scheme reveals relatively slight changes in ozone, hydroxyl radical, and nitrogen oxides on a global mean basis (1–4%), although remarkable regional differences (5–20%) exist near the source regions. $NO_x$ decreases over the source regions and increases in the remote troposphere, due mainly to more efficient transport of peroxyacetyl nitrate (PAN), which is increased with the SAPRC aromatic chemistry. Model ozone mixing ratios with the updated aromatic chemistry increase by up to 5 ppb (more than 10%), especially in industrially polluted regions. The ozone change is partly due to the direct influence of aromatic oxidation products on ozone production rates, and in part to the altered spatial distribution of $NO_x$ that enhances the tropospheric ozone production efficiency. Improved representation of aromatics is important to simulate the tropospheric oxidation.

## 1. Introduction

Non-methane volatile organic compounds (NMVOCs) play important roles in the tropospheric chemistry, especially in ozone production (Atkinson, 2000; Seinfeld and Pandis, 2012). Aromatic hydrocarbons such as benzene ($C_6H_6$), toluene ($C_7H_8$) and xylenes ($C_8H_{10}$) make up a large fraction of NMVOCs (Ran et al., 2009; Guo et al., 2006; You et al., 2008) in the atmosphere of urban and semi-urban areas. They are important precursors of secondary organic aerosol (SOA), peroxyacetyl nitrate (PAN), and ozone (Kansal, 2009; Tan et al., 2012; Porter et al., 2017). In addition, many aromatic compounds can cause detrimental effects on human health and plants (Manuela et al., 2012; Sarigiannis and Gotti, 2008; Michalowicz and Duda, 2007).

Aromatics are released to the atmosphere by biomass burning as well as fossil fuel evaporation and burning (Cabrera-Perez et al., 2016; Na et al., 2004). The dominant oxidation pathway for aromatics is via reaction with hydroxyl radical (OH, the dominant atmospheric oxidant), followed by reaction with nitrate radical ($NO_3$) (Cabrera-Perez et al., 2016; and references therein). The corresponding aromatic oxidation products could be involved in many atmospheric chemical processes, which can affect OH recycling and the atmospheric oxidation capacity (Atkinson and Arey, 2003; Calvert et al., 2002; Bejan et al., 2006; Chen et al., 2011). A realistic model description of aromatic compounds is necessary to improve our understanding of their effects on the chemistry in the atmosphere. However, up to now few regional or global-scale chemical transport models (CTMs) include detailed aromatic chemistry (Lewis et al., 2013; Cabrera-Perez et al., 2016).

Despite the potentially important influence of aromatic compounds on global atmospheric chemistry, their effect on global tropospheric ozone formation in polluted urban areas is less analyzed with the model simulation. The main source and sink processes of tropospheric ozone are photochemical production and loss, respectively (Seinfeld and Pandis 2006; Monks et al., 2015; Yan et al., 2016). Observation-based approaches alone cannot provide a full picture of ozone-source attribution for the different NMVOCs. Such ozone-source relationships are needed to improve policymaking strategies to address hemispheric ozone pollution (Chandra et al., 2006). Numerical chemistry-transport models allow us to explore the importance of impacts from aromatics and to attribute observed changes in ozone concentrations to particular sources (Stevenson et al., 2006; Stevenson et al., 2013; Zhang et al., 2014). Current global CTMs reproduce much of the observed regional and seasonal variability in tropospheric ozone concentrations. However, some systematic biases can occur, most commonly an overestimation over the northern hemisphere (Fiore et al., 2009; Reidmiller et al., 2009; Yan et al., 2016, 2018a, b; Ni et al., 2018) due to incomplete representation of physical and chemical processes, and biases in emissions and transport, including the parameterization of small-scale processes and their feedbacks to global-scale chemistry (Chen et al., 2009; Krol et al., 2005; Yan et al., 2014; Yan et al., 2016).

Another motivation for the modeling comes from recent updates in halogen (bromine-chlorine) chemistry, which when implemented in GEOS-Chem, a global chemical transport model being used extensively for tropospheric chemistry and transport studies (Zhang and Wang, 2016; Yan et

al., 2014; Shen et al., 2015; Lin et al., 2016), decrease the global burden of ozone significantly (by 14%; 2–10 ppb in the troposphere) (Schmidt et al., 2017). This ozone burden decline is driven by decreased chemical ozone production due to halogen-driven nitrogen oxides ($NO_x$ = $NO$ + $NO_2$) loss; and the ozone decline lowers global mean tropospheric OH concentrations by 11%. Thus GEOS-Chem starts to exhibit low ozone biases compared to ozonesonde observations (Schmidt et al., 2017), particularly in the southern hemisphere, implying that some mechanisms (e.g., due to aromatics) are currently missing from the model.

A simplified aromatic oxidation mechanism has previously been employed in GEOS-Chem (e.g., Fischer et al., 2014; Hu et al., 2015), which is still used in the latest version v12.0.0. In that simplified treatment, oxidation of benzene (B), toluene (T), and xylene (X) by OH (Atkinson et al., 2000) is assumed to produce first-generation oxidation products ($xRO_2$, x = B, T, or X). And these products further react with hydrogen peroxide ($HO_2$) or nitric oxide (NO) to produce $LxRO_2y$ (y = H or N), passive tracers which are excluded from tropospheric chemistry. Thus in the presence of $NO_x$, the overall reaction is aromatic + OH + NO = inert tracer. While such a simplified treatment can suffice for budget analyses of the aromatic species themselves, it does not capture ozone production from aromatic oxidation products.

In this work, we update the aromatics chemistry in GEOS-Chem based on the SAPRC-11 mechanism, and use the updated model to analyze the global and regional scale chemical effects of the most abundant aromatics in the gas phase (benzene, toluene, xylenes) in the troposphere. Specifically, we focus on the impact on ozone formation (due to aromatics oxidation), as this is of great interest for urban areas and can be helpful for developing air pollution control strategies. Further targets are the changes to the $NO_x$ spatial distribution and OH recycling. Model results for aromatics and ozone mixing ratios are evaluated by comparison with observations from surface and aircraft campaigns in order to constrain model accuracy. Finally, we discuss the global effects of aromatics on tropospheric chemistry including ozone, $NO_x$ and $HO_x$ ($HO_x$ = OH + $HO_2$).

The rest of the paper is organized as follows. Section 2 describes the GEOS-Chem model setups, including the updates in aromatics chemical mechanism. A description of the observational datasets for aromatics and ozone is given in Sect. 3. Section 4 presents the model evaluation for aromatics based on the previously mentioned set of aircraft and surface observations, and evaluates modeled surface ozone with measurements from three networks. An analysis of the tropospheric impacts on ozone, $NO_x$, and OH, examining the difference between models results with simplified (as in the standard model setup) and with SAPRC-11 aromatic chemistry, is presented in Section 5. Section 6 concludes the present study.

**2. Model description and setup**

We use the GEOS-Chem CTM (version 9-02, available at http://geos-chem.org/) to interpret the importance of aromatics in tropospheric chemistry and ozone production. GEOS-Chem is a

global 3-D chemical transport model for a wide range of atmospheric composition problems. It is driven by meteorological data provided from the Goddard Earth Observing System (GEOS) of the NASA Global Modeling Assimilation Office (GMAO). A detailed description of the GEOS-Chem model is available at http://acmg.seas.harvard.edu/geos/geos_chem_narrative.html. Here, the model is run at a horizontal resolution of 2.5º long. x 2º lat. with a vertical grid containing 47 layers (including 10 layers of ~ 130 m thickness each below 850 hPa), as driven by the GEOS-5 assimilated meteorological fields. The chemistry time step is 0.5 h, while the transport time step is 15 min in the model. A non-local scheme implemented by Lin and McElroy (2010) is used for vertical mixing in the planetary boundary layer. Model convection adopts the Relaxed Arakawa-Schubert scheme (Rienecker et al., 2008). Stratospheric ozone production employs the Linoz scheme (McLinden et al., 2000). Dry deposition for aromatic compounds is implemented following the scheme by Hu et al. (2015), which uses a standard resistance-in-series model (Wesely, 1989) and Henry's law constants for benzene (0.18 M atm$^{-1}$), toluene (0.16 M atm$^{-1}$), and xylenes (0.15 M atm$^{-1}$) (Sander, 1999).

## 2.1 Emissions

For anthropogenic NMVOCs emission including aromatic compounds (benzene, toluene, and xylenes), here we use emission inventory from the RETRO (REanalysis of the TROpospheric chemical composition) (Schultz et al., 2007). The global anthropogenic RETRO (version 2; available at ftp://ftp.retro.enes.org/) inventory includes monthly emissions for 24 distinct chemical species during 1960–2000 with a resolution of 0.5° long. × 0.5° lat. (Schultz et al., 2007). It is implemented in GEOS-Chem by regridding to the model resolution (2.5° long. × 2.0° lat.). Emission factors in RETRO are calculated on account of economic and technological considerations. In order to estimate the time dependence of anthropogenic emissions, RETRO also incorporate behavioral aspects (Schultz et al., 2007). The implementation of the monthly RETRO emission inventory in GEOS-Chem is described by Hu et al. (2015), which linked the RETRO species into the corresponding model tracers. Here the model speciation of xylenes includes m-xylene, p-xylene, o-xylene and ethylbenzene (Hu et al., 2015). The most recent RETRO data (for 2000) is used for the GEOS-Chem model simulation and the calculated annual global anthropogenic NMVOCs are ~ 71 TgC. On a carbon basis, the global aromatics (benzene + toluene + xylenes) source accounts for ~ 23% (16 TgC) of the total anthropogenic NMVOCs. Figure 1 shows the spatial distribution of anthropogenic emissions for benzene, toluene, and xylenes, respectively. Anthropogenic benzene emissions in Asia (mainly over eastern China and India) are larger than those from other source regions (e.g., over the Europe and eastern US).

Global NO$_x$ anthropogenic emissions are taken from the EDGAR (Emission Database for Global Atmospheric Research) v4.2 inventory. The global inventory has been replaced by regional inventories in China (MEIC, base year: 2008), Asia (excluding China; INTEX-B, 2006), the US (NEI05, 2005), Mexico (BRAVO, 1999), Canada (CAC, 2005), and Europe (EMEP, 2005). Details on these inventories and on the model NO$_x$ anthropogenic emissions are shown in Yan et al. (2016).

Biomass burning emissions of aromatics and other chemical species (e.g., $NO_x$) in GEOS-Chem are calculated based on the monthly Global Fire Emission Database version 3 (GFED3) inventory (van der Werf et al., 2010). Natural emissions of $NO_x$ (by lightning and soil) and of biogenic NMVOCs are calculated online by parameterizations driven by model meteorology. Lightning $NO_x$ emissions are parameterized based on cloud top heights (Price and Rind, 1992), and are further constrained by the lightning flash counts detected from satellite instruments (Murray et al., 2012). Soil $NO_x$ emissions are described in Hudman et al. (2012). Biogenic emissions of NMVOCs are calculated by MEGAN (Model of Emissions of Gases and Aerosols from Nature) v2.1 with the Hybrid algorithm (Guenther et al., 2012).

**2.2 Updated aromatic chemistry**

In the GEOS-Chem model setup, the current standard chemical mechanism with simplified aromatic oxidation chemistry is based on Mao et al. (2013), which is still the case for the latest version v12.0.0. As mentioned in the introduction, this simplified mechanism acts as strong sinks of both $HO_x$ and $NO_x$, because no $HO_x$ are regenerated in this reaction, and NO is consumed without regenerating $NO_2$. However, it is reasonably well established that aromatics tend to be radical sources, forming highly reactive products that photolyze to form new radicals, and regenerating radicals in their initial reactions (Carter, 2010a, b; Carter and Heo, 2013). A revised mechanism that takes the general features of aromatics mechanisms into account would be much more reactive, given the reactivity of the aromatic products.

This work uses a more detailed and comprehensive aromatics oxidation mechanism: the State-wide Air Pollution Research Center version 11 (SAPRC-11) aromatics chemical mechanism. SAPRC-11 is an updated version of the SAPRC-07 mechanism (Carter and Heo, 2013) to give better simulations of recent environmental chamber experiments. The SAPRC-07 mechanism underpredicted NO oxidation and $O_3$ formation rates observed in recent aromatic-$NO_x$ environmental chamber experiments. The new aromatics mechanism, designated SAPRC-11, is able to reproduce the ozone formation from aromatic oxidation that is observed in almost all environmental chamber experiments, except for higher (>100 ppb) $NO_x$ (Carter and Heo, 2013). Table S1 lists new model species in addition to those in the standard GEOS-Chem model setup. Table S2 lists the new reactions and rate constants. In this mechanism, the tropospheric consumption process of aromatics is mainly reaction with OH.

As discussed by Carter (2010a, b), aromatic oxidation has two possible OH reaction pathways: OH radical addition and H-atom abstraction (Atkinson, 2000). In SAPRC-11, taking toluene as an example in Table S2, the reactions following abstraction lead to three different formation products: an aromatic aldehyde (represented as the *BALD* species in the model), a ketone (*PROD2*), and an aldehyde (*RCHO*). The largest yield of toluene oxidation is the reaction after OH addition of aromatic rings. The OH-aromatic adduct is reaction with $O_2$ to form an OH-aromatic-$O_2$ adduct or $HO_2$ and a phenolic compound (further consumed by reactions with OH and $NO_3$ radicals). The OH-aromatic-$O_2$ adduct further undergos two competing unimolecular reactions to ultimately form OH, $HO_2$, an α-dicarbonyl (such as glyoxal (*GLY*), methylglyoxal

(*MGLY*) or biacetyl (*BACL*)), a monounsaturated dicarbonyl co-product (*AFG1*, *AFG2*, the photoreactive products) and a di-unsaturated dicarbonyl product (*AFG3*, the non-photoreactive products) (Calvert et al., 2002).

Formed from the phenolic products, the SAPRC-11 mechanism includes species of cresols (*CRES*), phenol (*PHEN*), xylenols and alkyl phenols (*XYNL*), and catechols (*CATL*). Due to their different SOA and ozone formation potentials (Carter et al, 2012), these phenolic species are represented separately. Relatively high yields of catechol (*CATL*) have been observed in the reactions of OH radicals with phenolic compounds. Furthermore, their subsequent reactions are believed to be important for SOA and ozone formation (Carter et al, 2012).

**2.3 Simulation setups**

In order to investigate the global chemical effects of the most commonly emitted aromatics in the troposphere, two simulations were performed, one with the ozone related aromatic chemistry updates from SAPRC-11 (the SAPRC case), and the other with simplified aromatic chemistry as in the standard setup (the Base case). Both simulations (Base and SAPRC) at 2.5° long. × 2° lat. are conducted from July 2004 to December 2005, allowing for a 6-month spin-up for our focused analysis over the year of 2005 for comparison to the available observations (Sect. 3). Initial conditions of chemicals are regridded from a simulation at 5° long. × 4° lat. started from 2004 with another spin-up run from January to June 2004. For comparison with aromatics observations over the US in 2010–2011 (Sect. 3), we extend the simulations from July 2009 to December 2011 with July-December 2009 as the spin-up period.

**3. Aromatics and ozone observations**

We use a set of measurements from surface and aircraft campaigns to evaluate the model simulated aromatics and ozone.

**3.1 Aromatic aircraft observations**

For aromatics, we use airborne observations from CALNEX (California; May/June 2010) aircraft study. A proton transfer reaction quadrupole mass spectrometer (PTR-MS) was used to measure mixing ratios of aromatics (and an array of other primary and secondary pollutants) during CALNEX. Measurements are gathered mostly on a one-second time scale (approximately 100 m spatial resolution), which permits sampling of the source regions and tracking subsequent transport and transformation throughout California and surrounding regions. Further details of the CALNEX campaign, including the flight track, timeframe, location and instrument, are shown in Hu et al. (2015) and https://www.esrl.noaa.gov/csd/projects/calnex. For comparison to the model results, we averaged the high temporal-spatial resolution observations to the model resolution.

We also employ vertical profiles obtained in 2005 from the CARIBIC (Civil Aircraft for Regular Investigation of the atmosphere Based on an Instrument Container) project, which conducts

atmospheric measurements onboard a commercial aircraft (Lufthansa A340-600) (Brenninkmeijer et al., 2007; Baker et al., 2010). CARIBIC flights fly away from Frankfurt, Germany on the way to North America, South America, India and East Asia. Measurements are available in the upper troposphere (50% on average) and lower stratosphere (50%) (UTLS) at altitudes between 10–12 km. To evaluate our results, measurements are averaged to the model output resolution. Vertically, results from GEOS-Chem model simulations at the 250 hPa level are used to compare with observations between 200–300 hPa. Then the annual means of observations and model data sampled along the flight tracks are used in the comparison.

## 3.2 Aromatics surface measurements

To evaluate the ground-level mixing ratios of benzene, toluene, and xylenes as well as their seasonal cycles, surface observations of aromatics are collected from two networks (EMEP, data available at http://www.nilu.no/projects/ccc/emepdata.html, and the European Environmental Agency (EEA), data available at http://www.eea.europa.eu/data-and-maps/data/airbase-the-european-air-quality-database-8, both for the year 2005) over Europe and the KCMP tall tower dataset (data available at https://atmoschem.umn.edu/data, for the year 2011) over the US.

EMEP, which aims to investigate the long-range transport of air pollution and the flux through geographic boundaries (Torseth et al., 2012), locates measurement sites in locations where there are minimal local impacts, thus consequently the observations could represent the feature of large regions. EMEP has a daily resolution with a total of 14 stations located in Europe for benzene, 12 stations for toluene, and 8 stations for xylenes (Table 1). Here we use the monthly values calculated from the database to evaluate monthly model results. Note that measurement speciation of xylenes (o-xylene, m-xylene and p-xylene) in EMEP network does not exactly correspond with the model speciation of xylenes (m-xylene, p-xylene, o-xylene and ethylbenzene) (Hu et al., 2015). The speciation assumption probably can partly account for the xylene model-measurement discrepancy seen in Sect. 4.

EEA provides observations from a large number of sites over urban, suburban and background regions (EEA, 2014). However, here we use only rural background sites to do model comparison, as in Cabrera-Perez et al. (2016), because the model horizontal scale cannot simulate direct traffic or industrial influence. This leads to 22 stations available for benzene and 6 stations for toluene. Further details of the sites and location information of EEA (and EMEP) used here are described in Cabrera-Perez et al., 2016. For comparison, annual means for individual sites have been used.

The KCMP tall tower measurements (at 44.69°N, 93.07°W, Minnesota, US) have been widely used for studies of surface fluxes of tropospheric trace species and land-atmosphere interactions (Kim et al., 2013; Hu et al., 2015; Chen et al., 2018). A suite of NMVOCs including aromatics were observed at the KCMP tower during 2009–2012 with a high-sensitivity PTR-MS, sampling from a height of 185 m above ground level. We averaged the hourly observations of benzene, toluene and $C_8$ (xylenes + ethylbenzene; here consistent with the model speciation) aromatics to

1 monthly values and then used for our model evaluation. Monthly mean simulations at the 990

2 hPa level (~190 m) are used for comparison.

## 3.3 Ozone observations

Ozone observations are taken from the database of the World Data Centre for Greenhouse Gases

(WDCGG, data available at http://ds.data.jma.go.jp/gmd/wdcgg/cgi-bin/wdcgg/catalogue.cgi),

and the Chemical Coordination Centre of EMEP (EMEP CCC). These networks contain hourly

ozone measurements over a total of 194 background sites in remote environments. We use

monthly averaged observations of surface ozone in 2005 to examine the simulated surface ozone

from the GEOS-Chem model. Simulated ozone from the lowest layer (centered at ~ 65 m) is

sampled from the grid cells corresponding to the ground sites.

## 4. Evaluation of simulated aromatics and ozone

In this section, the SAPRC model simulation results of aromatics (benzene, toluene, xylenes and

$C_8$ aromatics) and ozone from GEOS-Chem are evaluated with observations. Table 1 summarizes

the statistical comparison between measured and simulated concentrations over the monitoring

stations described in Sect. 3. For the statistical calculations, GEOS-Chem simulation results have

been sampled along the geographical locations of the measurements. Table 1 includes the number

of locations and time resolutions. The number of sites in EEA for xylenes is only 2, thus we do

not include their comparison results in Table 1 due to the lack of representativeness.

## 4.1 Surface-level aromatics

For the aromatics near the surface mixing ratios over Europe, observed mean benzene (194.0 ppt

for EEA and 166.4 ppt for EMEP) and toluene (240.3 ppt for EEA and 133.1 ppt for EMEP)

mixing ratios are higher than observed mean xylene concentrations (42.3 ppt for EMEP). In

general, the model underestimates EEA and EMEP observations of benzene (by 34% on average)

and toluene (by 20% on average). For benzene, the model results systematically underestimate

the annual means (36%) compared to the EMEP database, consistent with the model

underestimate of the EEA dataset (32%). The model underestimate for toluene compared to the

EMEP dataset (15%) is smaller than that relative to the EEA measurements (25%). The

simulation overestimates the xylene measurements in EMEP by a factor of 1.9, in part because

the model results include ethylbenzene but the observations do not (see Sect. 3.2). The fact that

the anthropogenic RETRO emissions (for year 2000) do not correspond to the year of

measurement (2005) may contribute to the above model-measurement discrepancies.

Anthropogenic aromatics emissions are reported to have significant changes in emissions and

their distributions over the decade by EDGARv4.3.2 (Crippa et al., 2018; http://eccad.aeris-

data.fr/#DatasetPlace:EDGARv4.3.2$DOI). It shows that the total aromatics emission from

anthropogenic source are enhanced by 5% (2005) and 14% (2011) compared to the year 2000.

The model bias would be partly benefit from this emission increase with enhanced modeled mixing ratios of benzene and toluene.

The modeled spatial variability of aromatics (with standard deviations of 32.1–66.8 ppt) is 18–73% lower than that of the EMEP and EEA observations (41.9–118.4 ppt), probably due to the coarse model resolution. The spatial variability in benzene (46–73% lower) is the most strongly underestimated among the three aromatic species. Unlike benzene, simulated concentrations of toluene show a larger standard deviation (66.8 ppt) than the EEA measurements (59.4 ppt), indicating larger simulated spatial variability. Simulation results are thus poorly spatially correlated with observations (R = 0.41–0.49). However, the temporal variability of aromatics is well captured by GEOS-Chem with the correlations above 0.7 for most stations.

Figure 2 shows a comparison of model results with observations at six stations for benzene, toluene, and xylenes, respectively, following Cabrera-Perez et al. (2016). The sites are chosen as the first six stations with largest amount of data. Model results reproduce the annual cycle at the majority of sites. Aromatics are better simulated in summer than in winter. This feature has been previously found for the climate-chemistry model EMAC for aromatics (Cabrera-Perez et al., 2016) and simpler NMVOCs (Pozzer et al., 2007). In addition, the measurements show larger standard deviations than the GEOS-Chem simulations, with the ratios between the observed and the simulated standard deviations being 2–11.

Over the US, annual mean observed concentrations at the KCMP tall tower are 91.5 ppt for benzene, 56.7 ppt for toluene, and 90.3 ppt for $C_8$ aromatics (Table 1). The model biases for benzene (8.4 ppt; 9.2%) and $C_8$ aromatics (−1.4 ppt; −1.6%) are much lower than that for toluene (64.5 ppt; 114%). Figure 3 further shows the observed and simulated monthly averaged concentrations of benzene, toluene and $C_8$ aromatics. The SAPRC simulation reproduces their seasonal cycles, with higher concentrations in winter and lower mixing ratios in summer, consistent with Hu et al. (2015). The model-observation correlations are 0.89, 0.78 and 0.65 for monthly benzene, toluene, and $C_8$ aromatics, respectively. The large overestimation of modeled toluene is mainly due to simulated high mixing ratios during the cold season (Fig. 3, October to March).

**4.2 Tropospheric aromatics**

Table 1 shows that in the UTLS, both CARIBIC observed (16 ppt) and GEOS-Chem modeled (12.3 ppt) benzene mixing ratios are higher than toluene concentrations (3.6 ppt for CARIBIC and 1.5 ppt for GEOS-Chem). For benzene, the model underestimates appear to be smaller in the free troposphere (with an underestimate by 23%) than at the surface (36% for EMEP and 32% for EEA). In contrast to benzene, annual mean concentrations of toluene are underestimated by 58% in the UTLS. The geographical variability of benzene is larger than that for toluene (with standard deviation of 4.2 versus 0.7 ppt in model and 15.8 versus 7.5 ppt in observation), probably because of the shorter lifetime of benzene (between several hours and several days;

http://www.nzdl.org/gsdlmod?a=p&p=home&l=en&w=utf-8) in combination with the lower
concentrations in the UTLS for toluene. The model results show smaller spatial variability than
the observations. This underestimation for spatial variability in the free troposphere (over 70%) is
higher than that at the surface (not shown).

The black lines in Fig. 4 show the tropospheric aromatics profiles during the CALNEX
campaign. The measured values peak at an altitude of 0.6–0.8 km, with concentrations decreasing
at higher altitudes. Although the concentrations in the lower troposphere for benzene (40–100 ppt
below 2 km) are lower than mixing ratios for toluene (70–160 ppt below 2 km) and $C_8$ aromatics
(50–120 ppt below 2 km), the benzene mixing ratios (> 30 ppt) in the free troposphere are much
higher than those of toluene and $C_8$ aromatics (< 10 ppt). The different profile shapes in the lower
troposphere for benzene, toluene and $C_8$ aromatics are mainly due to their different emissions and
lifetime. The SAPRC simulation (red lines in Fig. 4) captures the general vertical variations of
CALNEX benzene and toluene, with statistically significant model-observation correlations of
0.74 and 0.65 for benzene and toluene, respectively. The model generally overestimates the
measured $C_8$ aromatics below 0.5 km, albeit with an underestimate above 0.5 km, with lower
model-observation correlation of 0.37. This overestimation below 0.5 km is also seen for benzene
and toluene. The modeled overly rapid aromatics drop-off with altitude probably implies the
modelled aromatics lifetime is short.

**4.3 Surface ozone**

Table 1 shows an average ozone mixing ratio of 34.1 ppb in 2005 over the regional background
WDCGG sites. The annual mean ozone mixing ratios are lower over Europe (from the EMEP
dataset), about 30.6 ppb. The SAPRC simulation tends to underestimate the mixing ratios over
the sites of Europe and background regions with biases of −2.9 ppb and −5.5 ppb, respectively.
Figure 5 shows the spatial distribution of the annual mean model biases with respect to the
measurements. Unlike the modeled surface aromatics, the simulated ozone spatial variability can
be either slightly lower or higher than the observed variability, depending on the compared
database: the standard deviation is 12.8 ppb (simulated) versus 14.2 ppb (observed) for WDCGG
sites, 13.2 versus 10.3 ppb for EMEP sites. The temporal variability (temporal correlations of
0.68–0.72) is better captured by the model than the spatial variability (spatial correlations of
0.52–0.54).

**5. Global effects of aromatic chemistry**

This section compares the Base and SAPRC simulations to assess to which extent the updated
mechanism for aromatics affect the global simulation of ozone, $HO_x$ and individual nitrogen
species. Our focus here is on the large-scale impacts.

**5.1 $NO_y$ Species**

Figure 6 and Table 2 show the changes from Base to SAPRC in annual average surface NO mixing ratios. A decrease in NO is apparent over $NO_x$ source regions, e.g., by approximately 0.15 ppb (~20%) over much of the US, Europe and China (Fig. 6). In contrast, surface NO increases at locations downwind from $NO_x$ source regions (up to ~0.1 ppb or 20%), including the oceanic area off the eastern US coast, the marine area adjacent to Japan, and the Mediterranean area. The change is negligible (by −0.2%) for the annual global mean surface NO (Table 2). Seasonally, the decrease in spring, summer and fall is compensated partly by the increase in winter (Table 2). This winter increase versus decline in other seasons is probably attributed to the weaken photochemical reactions involving $NO_x$ in winter.

The zonal average results in Fig. 7 show a clear decline in NO in the planetary boundary layer, in contrast to significant increases in the free troposphere, from Base to SAPRC. The free tropospheric NO increases are about the same from 30°S-90°N with an annual average enhancement up to 5% (Fig. 7), and are particularly large in winter (up to 10%, not shown). For the whole troposphere, the average NO increases by 0.6% from Base to SAPRC (Table 2).

Figure 6 shows that simulated surface $NO_2$ mixing ratios in the SAPRC scenario are enhanced over most locations across the globe, in comparison with the Base simulation. Over the source regions, the changes are mixed, with increases in some highly $NO_x$ polluted regions (by up to 10%) and decreases in other polluted regions. On a global mean basis, $NO_2$ is increased (by 2.1% in the free troposphere and 1.0% at the surface, Table 2), due mainly to the recycling of $NO_x$ from PAN associated with the aromatics, and the reactions of oxidation products from aromatics with NO or $NO_3$ (primarily) to form $NO_2$ and $HO_2$. Combing the changes in NO and $NO_2$ means that the total $NO_x$ mixing ratios decrease in source regions but increase in the remote free troposphere (Fig. 8 and 9).

The $NO_3$ mixing ratios decrease at the global scale (−4.1% on average in the troposphere, Fig. 7 and Table 2) in the SAPRC simulation, except for an enhancement in surface $NO_3$ over the northern polar regions and most polluted areas like the eastern US, Europe and eastern China (Fig. 6). The $NO_3$ global decreases are mainly due to the consumption of $NO_3$ by reaction with the aromatic oxidation products. However, the $NO_3$ regional increases are probably caused by the enhanced regional atmospheric oxidation capacity.

Table 2 shows that nitric acid ($HNO_3$) increases in the SAPRC simulation, both near the surface (by approximately 1.1%) and in the troposphere (by 0.3%). The enhancement in $HNO_3$ appears uniformly over most continental regions in the northern hemisphere (not shown), due to the promotion of direct formation of $HNO_3$ from aromatics in the SAPRC simulation.

**5.2 OH and $HO_2$**

Compared to the Base simulation, OH increases slightly by 1.1% at the surface in the SAPRC simulation, with that declines over the tropics (30°S−30°N) are compensated by enhancements over other regions (Fig. 10 and Table 2). The largest increases in OH concentrations are found

over source regions dominated by anthropogenic emissions (i.e., the US, Europe, and Asia) and in subtropical continental regions with large biogenic aromatic emissions. In these locations, the peroxy radicals formed by aromatic oxidation react with NO and $HO_2$, which can have a significant effect on the ambient ozone and $NO_x$ mixing ratios. This in turn influences OH, as the largest photochemical sources of OH in the model are the photolysis of $O_3$ as well as the reaction of NO with $HO_2$. Seasonally, a few surface locations see OH concentration increases of more than 10% during April–August (not shown), including parts of the eastern US, central Europe, eastern Asia and Japan.

The OH enhancement (0.2%) is also seen in the free troposphere in the SAPRC simulation (Fig. 11 and Table 2). OH is increased in the troposphere of the northern hemisphere, in contrast to the decline in the troposphere of tropics and southern hemisphere (Fig. 11). These OH changes correspond to the hemispherically distinct changes in aromatics (benzene, toluene, and xylenes), which show a decrease in the northern hemisphere, an increase in the southern hemisphere (Fig. 12 and 13), and an increase in global mean (by 1%) (Table 2). Despite the overall increase in tropospheric OH, CO is increased by ~1% (Table 2) due to additional formation from aromatics oxidation.

Table 2 shows that from Base to SAPRC, $HO_2$ shows a significant increase at the global scale: 3.0% at the surface and 1.3% in the troposphere, due to regeneration of $HO_x$ from aromatics oxidation products. Correspondingly, the $OH/HO_2$ ratio decreases slightly. These changes mean that, compared to the simplified aromatic chemistry in the standard model setup, the SAPRC mechanism are associated with higher OH (i.e., more chemically reactive troposphere) and even higher $HO_2$.

**5.3 Ozone**

From Base to SAPRC, the global average surface ozone mixing ratio increases by less than 1% (Table 2). This small difference is comparable to the result calculated by Cabrera-Perez et al. (2017) with the EMAC model, which is based on a reduced version of the aromatic chemistry from the Master Chemical Mechanism (MCMv3.2). Figure 10 shows that the 1% increase in surface ozone occurs generally over the northern hemisphere. Similar to the changes in OH, the most notable ozone increase occurs in industrially-polluted regions. These regions show significant local ozone photochemical formation in both the Base case and the SAPRC simulation. The updated aromatic chemistry increases ozone by up to 5 ppb in these regions. Increases of ozone are much smaller (less than 0.2 ppb) over the tropical oceans than in the continental areas. In contrast, ozone declines in regions of South America, Central Africa, Australia and Indonesia over the tropics (30°S–30°N). Changes elsewhere in the troposphere are similar in magnitude, as shown in Figure 11.

Two general factors likely contribute to the ozone change from Base to SAPRC. In the SAPRC simulation, the addition of aromatic oxidation products (i.e., peroxy radicals) can contribute

directly to ozone formation in $NO_x$-rich source regions and also in the $NO_x$-sensitive remote troposphere (i.e., from PAN to $NO_x$ and to ozone). The second factor is a change in the $NO_x$ spatial distribution, with an overall enhancement in average $NO_2$ concentrations. The redistribution is mainly caused by enhanced transport of $NO_x$ to the remote troposphere (see Sect. 5.1). The enhanced $NO_x$ in the remote troposphere enhances the overall ozone formation because this process is more efficient in the remote regions (e.g., Liu et al., 1987). The increased ozone, $NO_2$ and $NO_x$ transport all lead to the aforementioned changes. This is described in detail in section 5.4.

There are notable decreases (more than 5%, Fig. 11) in simulated ozone and OH in the free troposphere (above 4 km) over the tropics (30°S–30°N). A similar decrease is found in modeled $NO_x$ (above 6 km, Fig. 9). These decreases are probably related to the upward transport of aromatics by tropical convection processes. The aromatics transported to the upper troposphere may cause net consumption of tropospheric OH and $NO_x$, which can further reduce ozone production.

From Base to SAPRC, the modeled ozone concentrations are close to the WDCGG and EMEP network measurements (Table 3). For the WDCGG background sites, the annual and seasonal model biases are ~10% smaller in the SAPRC simulation compared to the Base case. For the EMEP stations, although the model results are not improved in summer and fall, the annual model bias is 25% smaller (−2.8 ppb versus −3.5 ppb) in the SAPRC simulation.

## 5.4 Discussion of SAPRC aromatic-ozone chemistry

As discussed in Sect. 5.3, the increased $O_3$ mixing ratios from Base to SAPRC are due to the direct impact of aromatic oxidation products (i.e., peroxy radicals) and to the effect of increased $NO_2$ concentrations. The simulated odd oxygen family ($O_x = O_3 + O(^1D) + O(^3P) + NO_2 + 2 \times NO_3 + 3 \times N_2O_5 + HNO_3 + HNO_4 + PAN$, Wu et al., 2007; Yan et al., 2016) formation increases by 1−10%, both over the source regions and in the remote troposphere (Fig. 10 and 11). Although the percentage changes are similar, the driving factors over the source regions are different from the drivers in the remote troposphere.

Regions with large aromatics emissions show a significant increase of oxidation products from Base to SAPRC. The modeled ozone in these regions increases with increasing $NO_2$ and its oxidation products. NO and $NO_3$ are often lower in these regions in the SAPRC scenario because of their reactions with the aromatic-OH oxidation products to form $NO_2$ and $HO_2$. In remote regions and in the free troposphere, ozone production is also enhanced by both $NO_2$ and $HO_2$ increases in the SAPRC simulation, but the increase in ozone formation is mainly attributed to the increase in $NO_x$ mixing ratios.

$NO_x$ concentrations decrease in source regions and increase in the remote regions because of more efficient transport of PAN and its analogues (represented by *PBZN* here in SAPRC-11). From Base to SAPRC, modeled PAN has been enhanced in a global scale (Fig. 8 and 9) via

reactions of aromatic-OH oxidation products with $NO_2$ (equation of BR13 in Table S2). In the SAPRC-11 aromatics chemical scheme the immediate precursor of PAN (peroxyacetyl radical) has five dominant photochemical precursors. They are acetone ($CH_3COCH_3$, model species: *ACET*), methacrolein (*MACR*), biacetyl (*BACL*), methyl glyoxal (*MGLY*) and other ketones (e.g., *PROD2*, *AFG1*). These compounds explain the increased rate of PAN formation. For example, the SAPRC simulation has increased the concentration of *MGLY* by a factor of 2. In addition, production of organic nitrates (*PBZN* (reactions of BR30 and BR31 in Table S2) and *RNO3* (PO36)) in the model with SAPRC aromatics chemistry may also explain the increase in ambient $NO_x$ in the remote regions, due to the re-release of $NO_x$ from organic nitrates (as opposed to removal by deposition). Due to such re-release of $NO_x$ from PAN-like compounds and also transport of $NO_x$, $NO_x$ increases by up to 5% at the surface in most remote regions and by ~1% in the troposphere as a whole. This then leads to increased ozone due to the effectiveness of ozone formation in the free troposphere.

SAPRC is a highly efficient and compact chemical mechanism with the use of maximum ozone formation as a primary metric in the chamber experiment benchmark. The mechanism has been primarily used and evaluated in regional CTMs such as CMAQ and CAMx, at much finer resolution (i.e., a few kilometers). Our study has significant application to use it in a global model. Implementing SAPRC-11 aromatic chemistry would add ~3% more computational effort in terms of model simulation times.

SAPRC is based on lumped chemistry, which is partly optimized on empirical fitting to smog chamber experiments that are representative to one-day photochemical smog episodes typical of, for example, Los Angeles and other US urban centers. However, SAPRC-11 gives better simulations of ozone formation in almost all conditions, except for higher (>100 ppb) $NO_x$ experiments where $O_3$ formation rates are consistently over predicted (Carter and Heo, 2013). This over prediction can be corrected if the aromatics mechanism is parameterized to include a new $NO_x$ dependence on photoreactive product yields, but that parameterization is not incorporated in SAPRC-11 because it is inconsistent with available laboratory data.

Other option, such as the condensed MCM mechanism, which are based upon more fundamental laboratory and theoretical data and used for policy and scientific modelling multi-day photochemical ozone formation, is experienced over Europe by Cabrera-Perez. (2016). Our results are consistent with the simulation of EMAC model implemented with a reduced version of the MCM aromatic chemistry. Moreover, aromatic chemistry is still far from being completely understood. For example, Bloss et al., (2005) show that for alkyl substituted mono-aromatics, when comparisons to chamber experiment over a range of VOC/$NO_x$ conditions, the chemistry under predicts the reactivity of the system but over predicts the amount of $O_3$ formation (model shows more NO to $NO_2$ conversion than on the experiments).

## 6. Conclusions

A representation of tropospheric reactions for aromatic hydrocarbons in the SAPRC-11 mechanism has been added to GEOS-Chem, to provide a more realistic representation of their atmospheric chemistry. The GEOS-Chem simulation with the SAPRC-11 aromatics mechanism has been evaluated against measurements from aircraft and surface campaigns. The comparison with observations shows reasonably good agreement for aromatics (benzene, toluene, and xylenes) and ozone. Model results for aromatics can reproduce the seasonal cycle, with a general underestimate over Europe for benzene and toluene, and an overestimate of xylenes; while over the US a positive model bias for benzene and toluene and a negative bias for $C_8$ aromatics are found. From the Base to the SAPRC simulation, the model ozone bias is reduced by 10% relative to WDCGG observations and by 25% relative to EMEP observations.

The simplified aromatics chemistry in the Base simulation under-predicts NO and $NO_3$ oxidation, and it does not represent ozone formed from aromatic-OH-$NO_x$ oxidation. Although the global average changes in simulated chemical species are relatively small (1%–4% from Base to SAPRC), on a regional scale the differences can be much larger, especially over aromatics and $NO_x$ source regions. From Base to SAPRC, $NO_2$ is enhanced by up to 10% over some highly polluted areas, while reductions are notable in other polluted areas. Although the simulated surface NO decreases by approximately 0.15 ppb (~20%) or more in the northern hemispheric source regions, including most of the US, Europe and China, increases are found (~0.1 ppb, up to 20%) at locations downwind from these source regions. The total $NO_x$ mixing ratios decrease in source regions but increase in the remote free troposphere. This is mainly due to the addition of aromatics oxidation products in the model that lead to PAN, which facilitates the transport of nitrogen oxides to downwind locations remote from the sources. Finally, the updated aromatic chemistry in GEOS-Chem increases ozone concentrations, especially over industrialized regions (up to 5 ppb, or more than 10%). Ozone changes in the model are partly explained by the direct impact of increased aromatic oxidation products (i.e., peroxy radical), and partly by the effect of the altered spatial distribution of $NO_x$. Overall, our results suggest that a better representation of aromatics chemistry is important to model the tropospheric oxidation capacity.

**Data Availability**

The aircraft and surface data used in this paper is already publically available. Airborne observations of aromatics from CALNEX (https://www.esrl.noaa.gov/csd/projects/calnex) and CARIBIC project. Surface observations of aromatics are collected from EMEP (http://www.nilu.no/projects/ccc/emepdata.html) and EEA (http://www.eea.europa.eu/data-and-maps/data/airbase-the-european-air-quality-database-8) over Europe and the KCMP tall tower dataset (https://atmoschem.umn.edu/data) over the US. Ozone observations are taken from WDCGG (http://ds.data.jma.go.jp/gmd/wdcgg/cgi-bin/wdcgg/catalogue.cgi).

**Code Availability**

The GEOS-Chem code of version 9-02 used to generate this paper and the model results are available upon request. We are submitting the code for inclusion into the standard model. The revised aromatics chemistry will be incorporated in the current version 12.0.0 and the later versions.

**Acknowledgements**

This research is supported by the National Natural Science Foundation of China (41775115), the 973 program (2014CB441303) and the Key Program of Ministry of Science and Technology of the People's Republic of China (2016YFA0602002; 2017YFC0212602). The research was also funded by the Start-up Foundation for Advanced Talents (162301182756). We acknowledge the free use of ozone data from networks of WDCGG (http://ds.data.jma.go.jp/gmd/wdcgg/cgi-bin/wdcgg/catalogue.cgi), EMEP (http://www.nilu.no/projects/ccc/emepdata.html) and aromatic compounds observations from EEA (http://www.eea.europa.eu/data-and-maps/data/airbase-the-european-air-quality-database-8) and EMEP. We also want to thank Angela Baker for providing the CARIBIC data. DBM acknowledges support from NASA (Grant #NNX14AP89G).

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

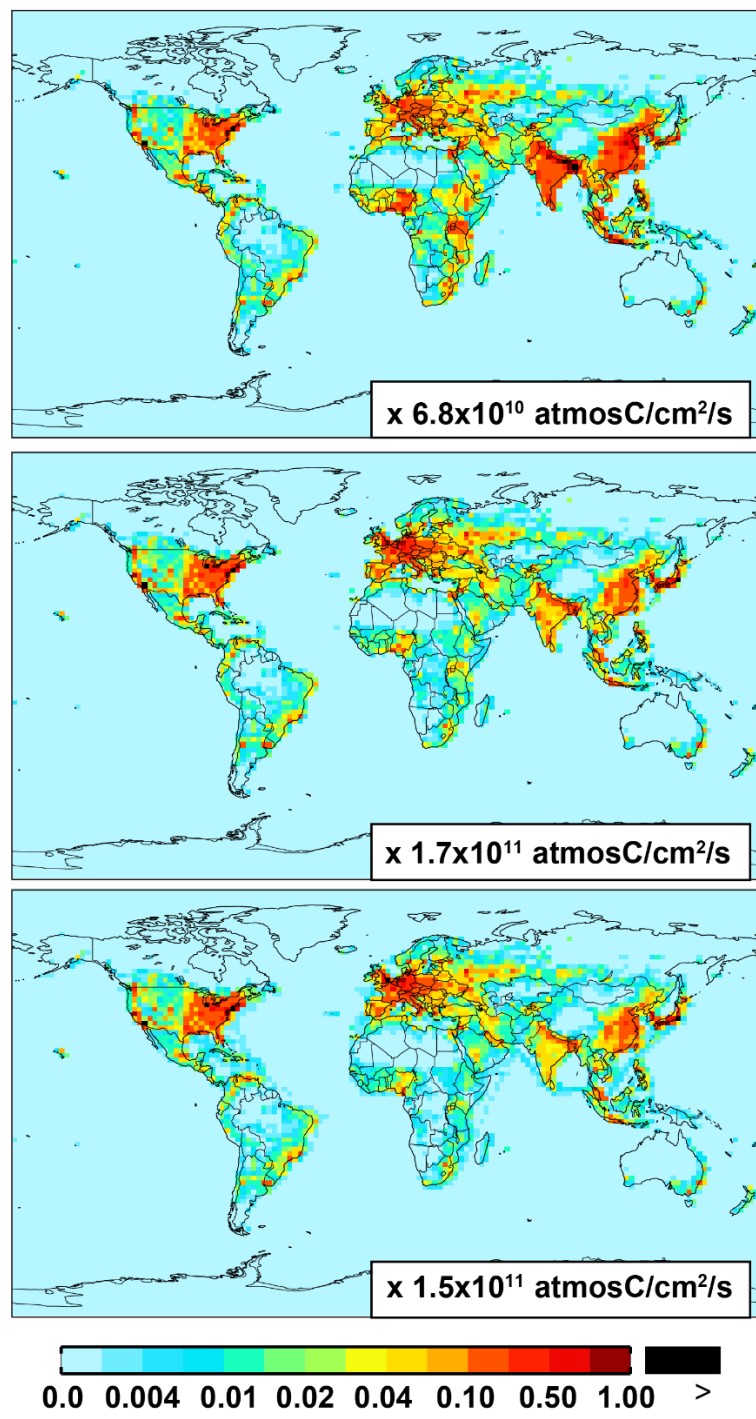

Figure 1. Spatial distribution of anthropogenic emissions from RETRO for benzene (top), toluene (middle), and xylenes (bottom), respectively.

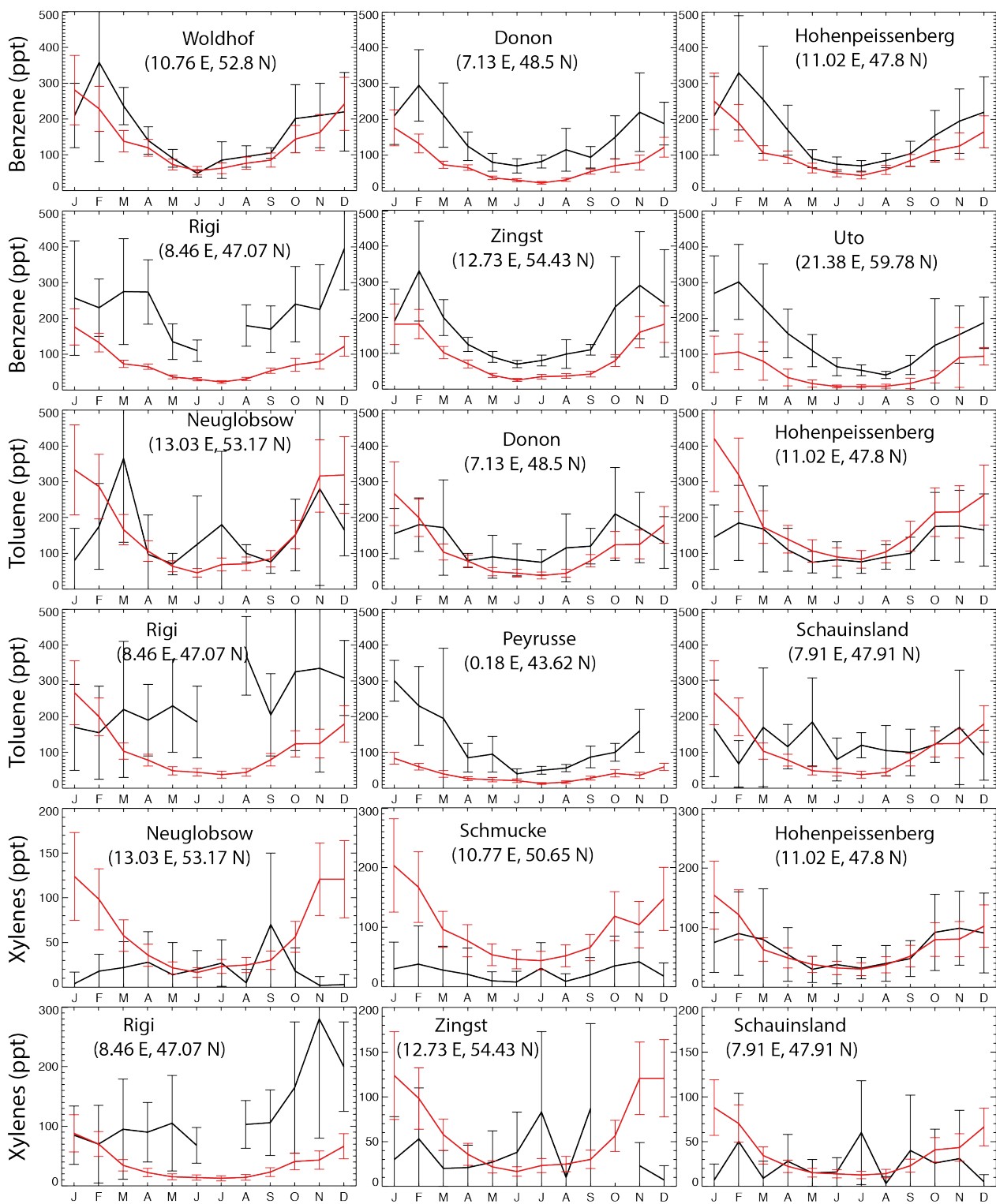

Figure 2. Monthly average EMEP observations (in black) of benzene (first two rows), toluene (middle two rows) and xylenes (last two rows) at six different locations for the year 2005, as well as the model results in the SAPRC simulation (in red), both in ppt. Error bars show the standard deviations.

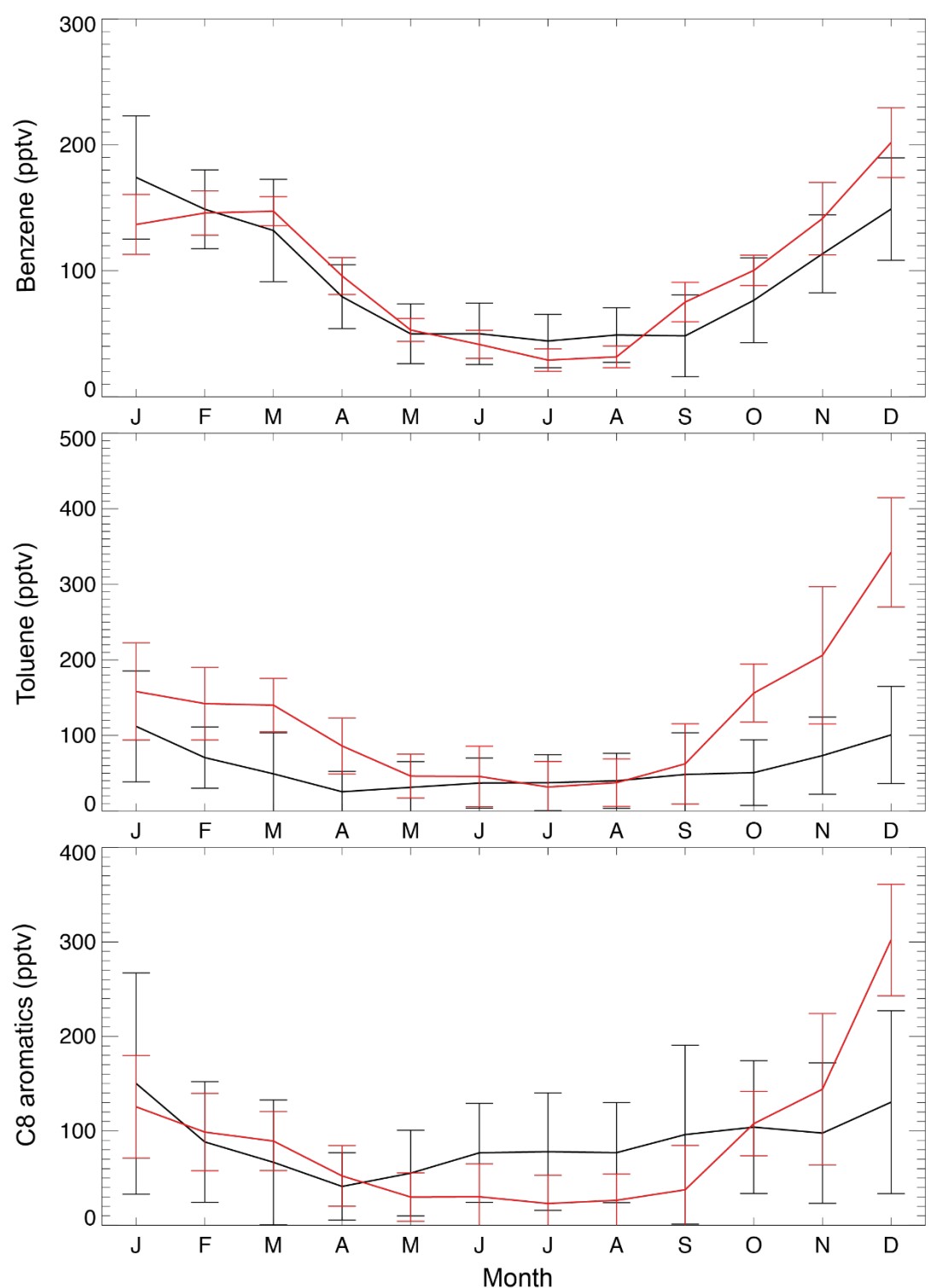

2    Figure 3. Monthly average KCMP tall tower observations (in black) of benzene, toluene and C$_8$ (xylenes +

3    ethylbenzene) aromatics in the year 2011 and the model results in the SAPRC simulation (in red). Error

4    bars show the standard deviations.

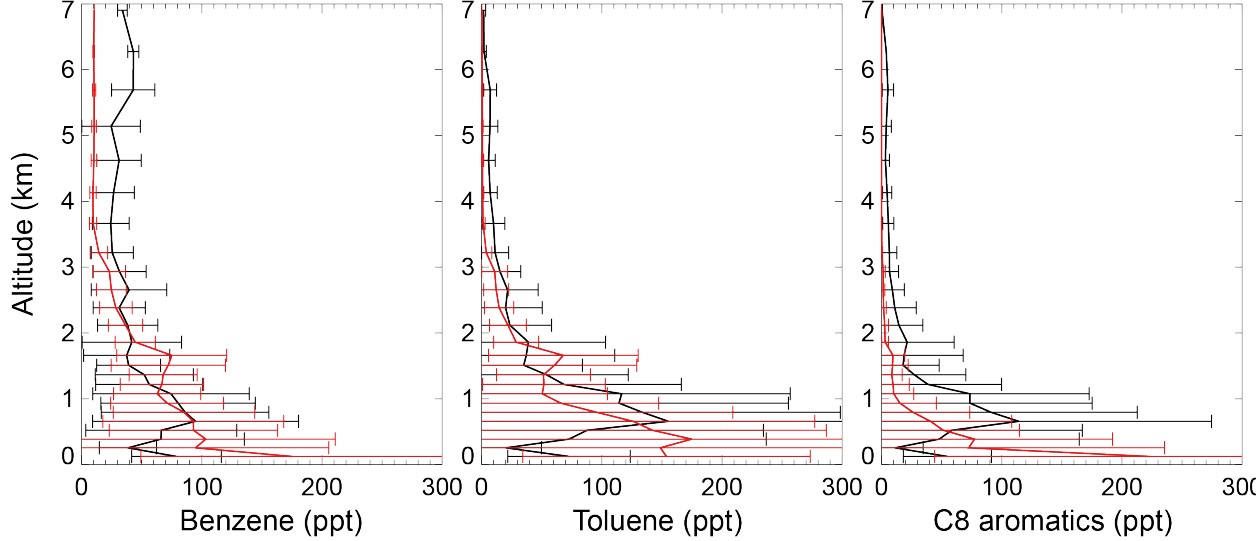

Figure 4. Measured (black) and simulated (red for the SAPRC case) vertical profiles of aromatics in May/June 2010 for the CALNEX campaigns. Model results are sampled at times and locations coincident to the measurements. Horizontal lines indicate the standard deviations.

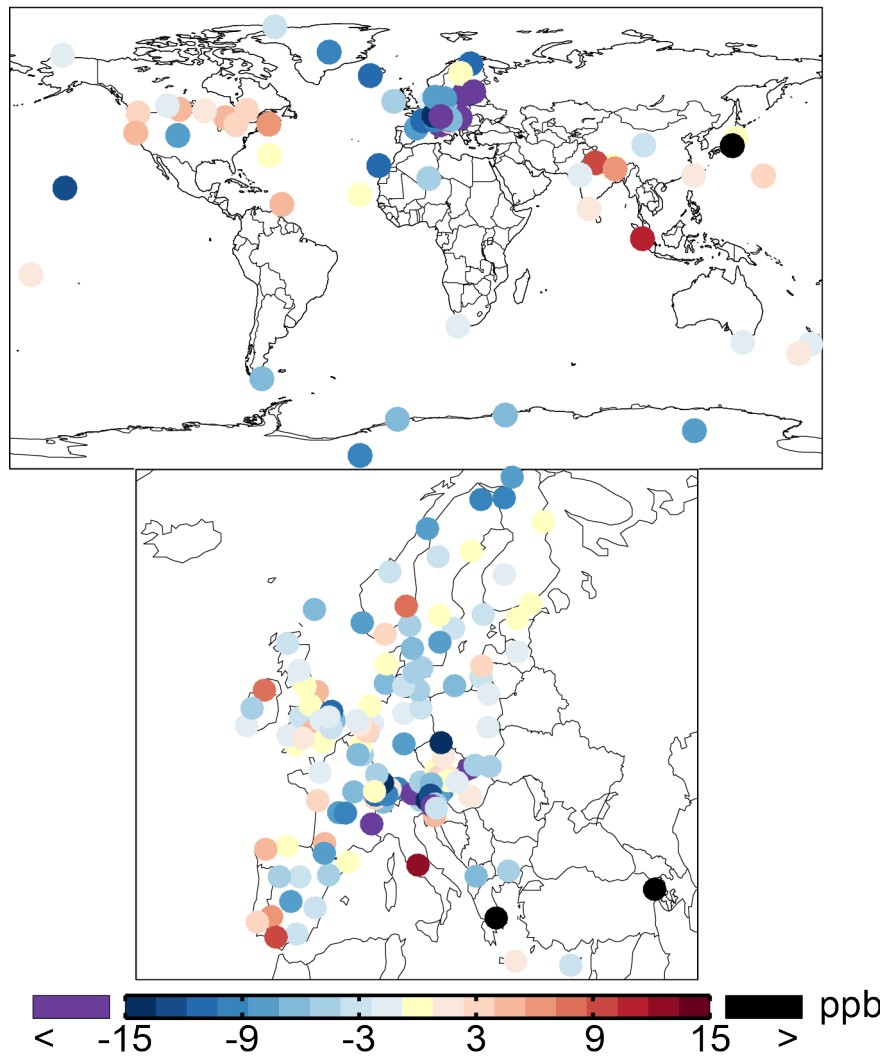

Figure 5. Annual mean model biases for surface ozone in the SAPRC simulation, with respect to measurements from WDCGG (top panel), and EMEP (bottom panel) networks.

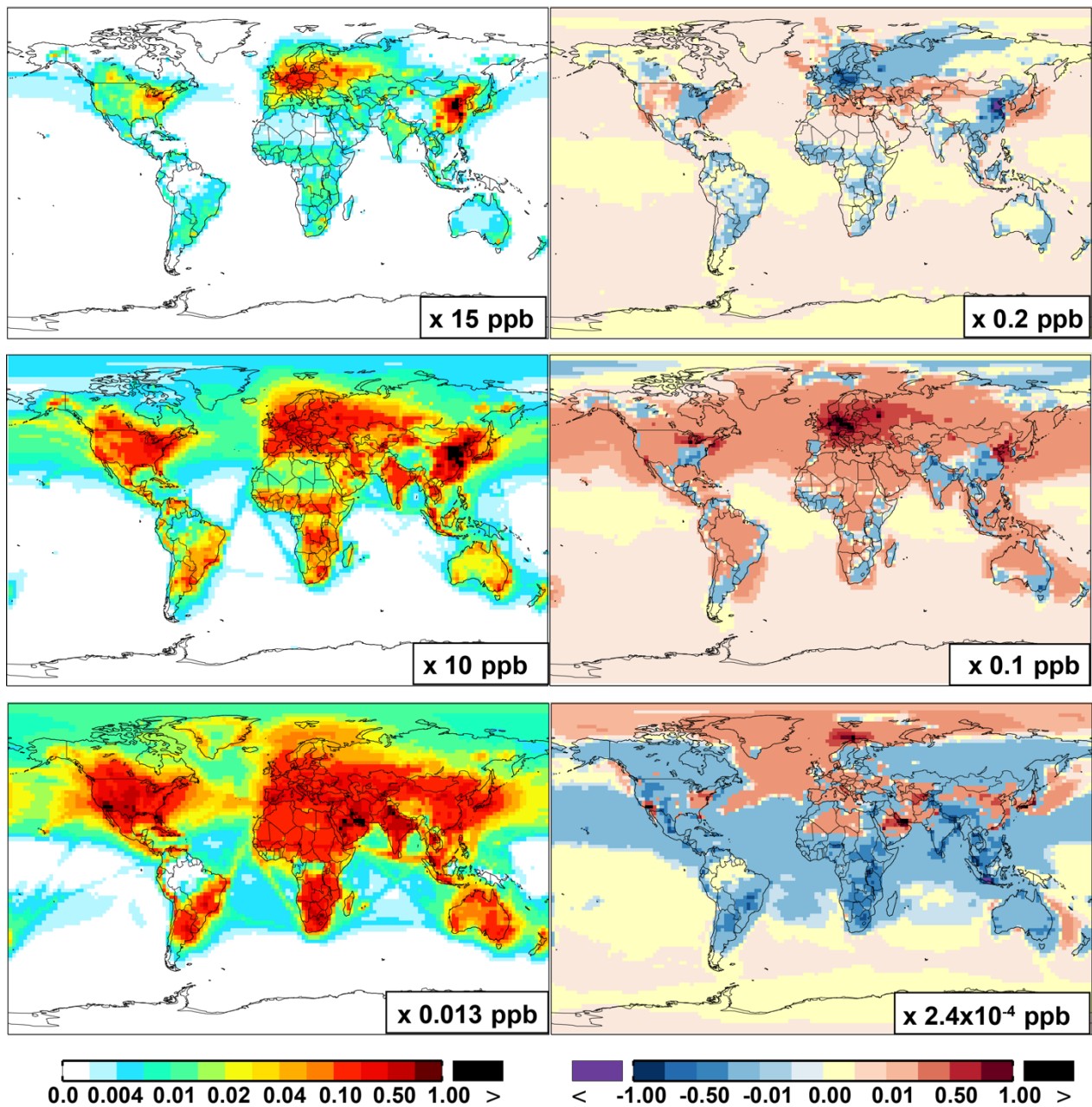

Figure 6. (Left column) Modeled spatial distributions of annual mean surface NO (top), $NO_2$ (middle), and $NO_3$ (bottom) simulated in the Base case for the year 2005. (Right column) The respective changes from Base to SAPRC.

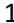

3  Figure 7. (Left column) Modeled zonal average latitude-altitude distributions of annual mean NO (top)
4  and $NO_2$ (middle), and $NO_3$ (bottom) simulated in the Base scenario for the year 2005. (Right column)
5  The respective changes from Base to SAPRC.

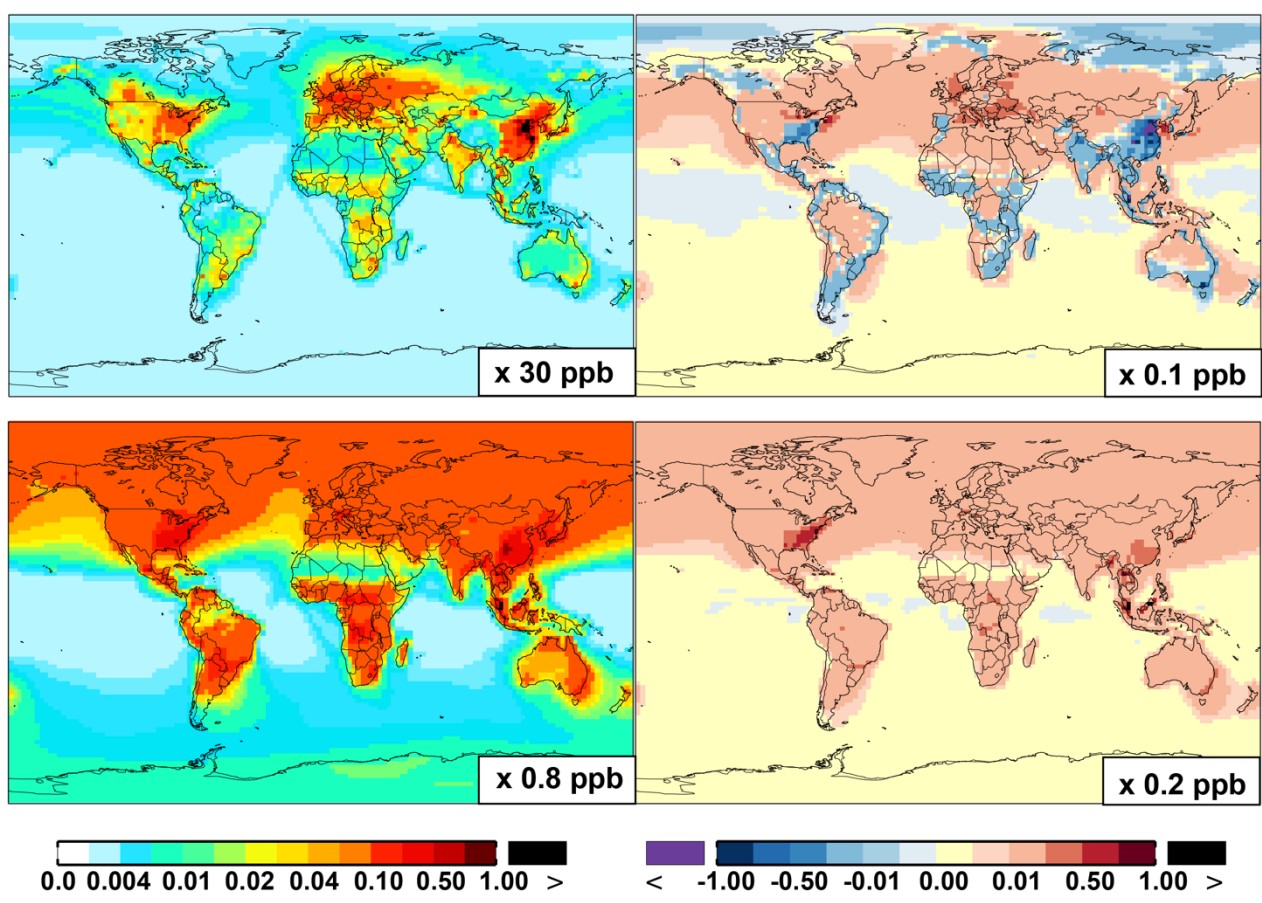

3  Figure 8. Same as Fig. 6 but for $NO_x$ (top panels) and PAN (bottom panels).

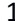

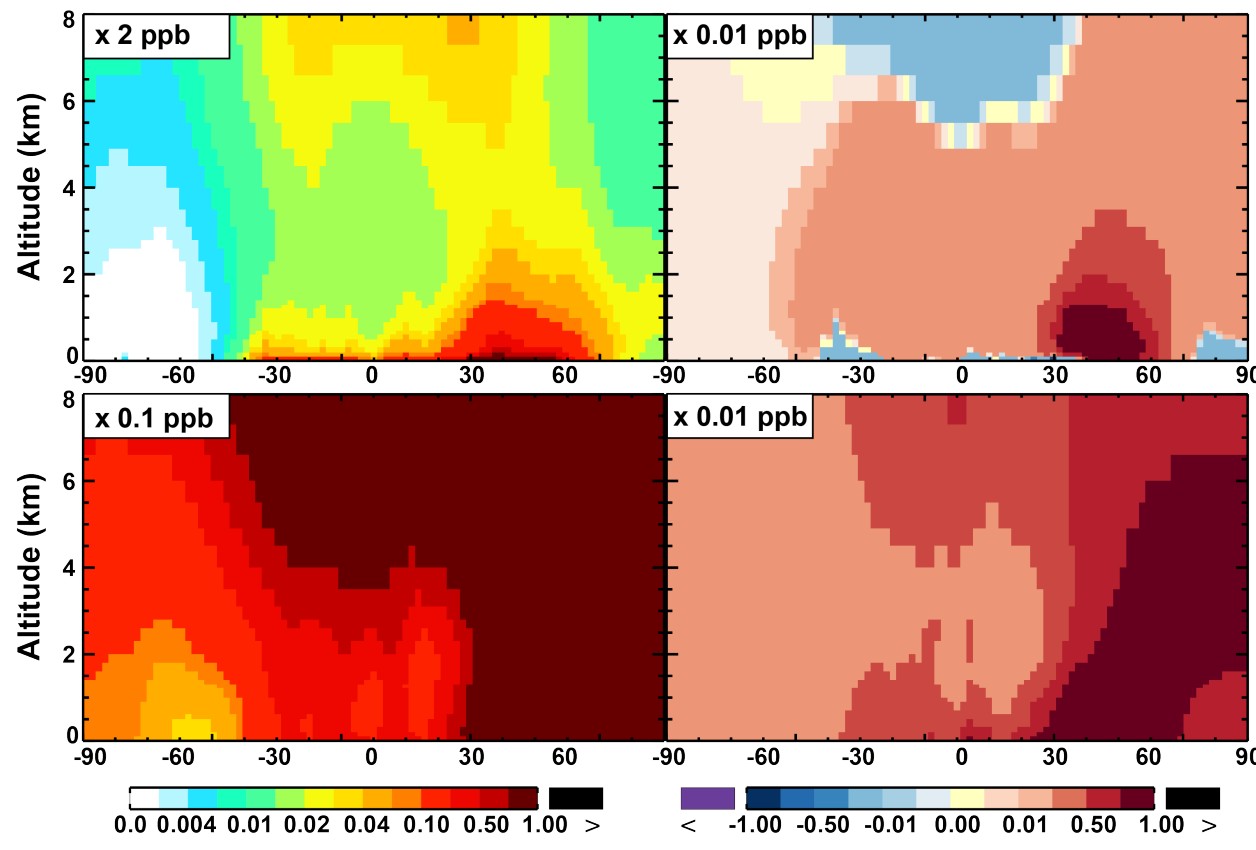

Figure 9. Same as Fig. 7 but for $NO_x$ (top panels) and PAN (bottom panels).

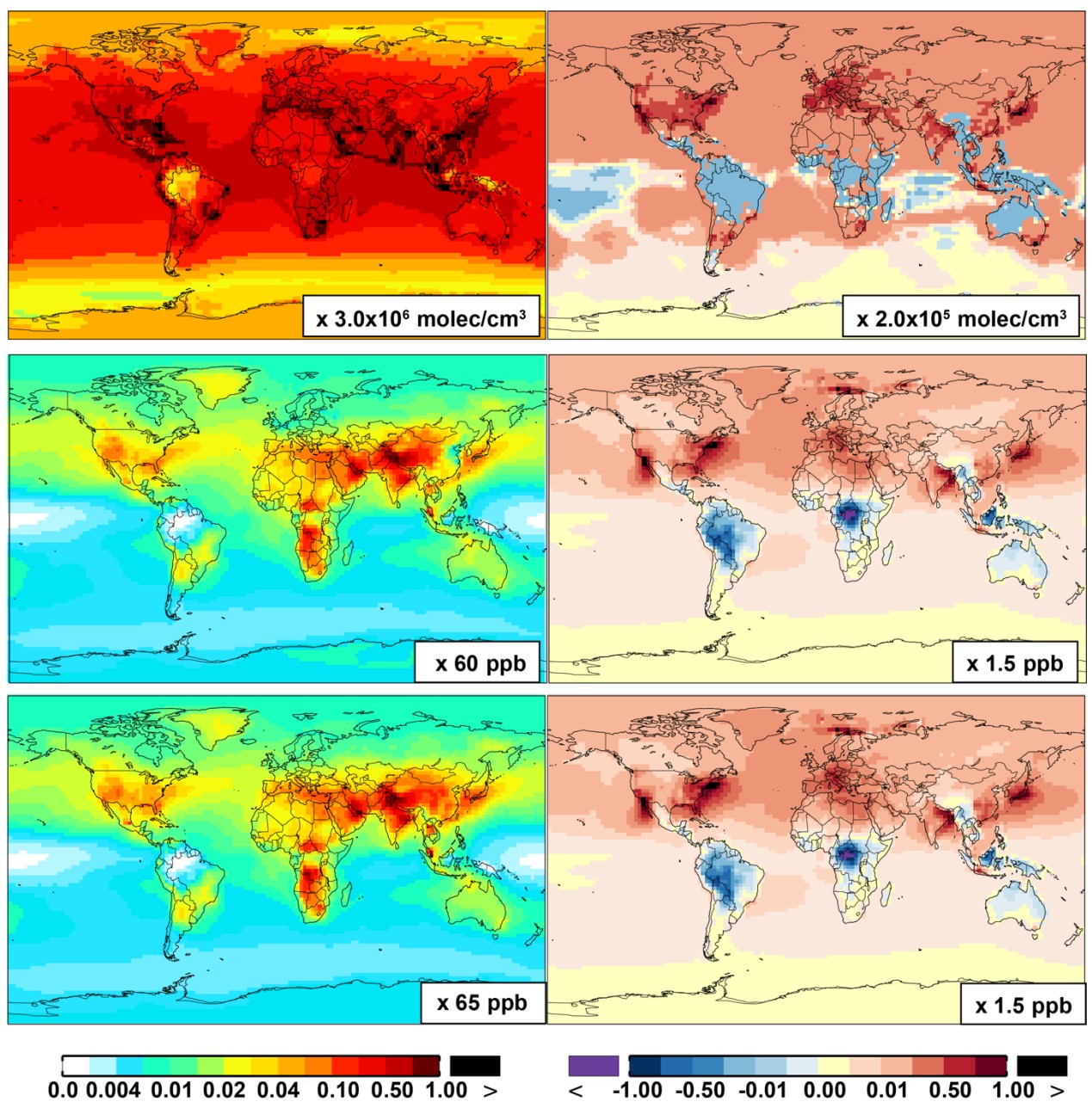

Figure 10. Same as Fig. 6 but for OH (top panels), $O_3$ (middle panels) and $O_x$ (bottom panels).

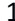

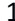

3    Figure 11. Same as Fig. 7 but for OH (top panels), $O_3$ (middle panels) and $O_x$ (bottom panels).

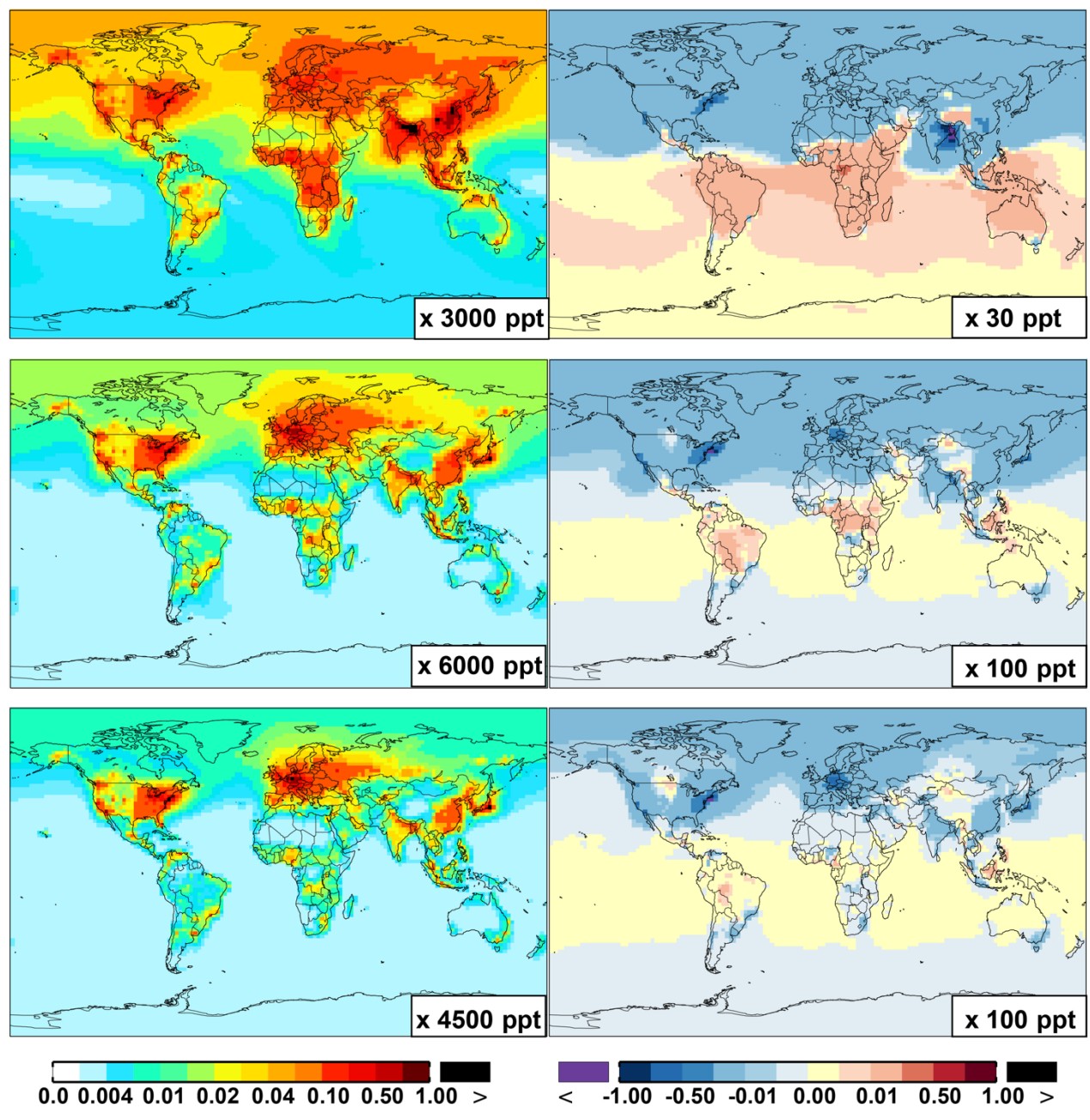

Figure 12. Same as Fig. 6 but for benzene (top panels), toluene (middle panels) and xylene (bottom panels).

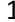

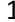

Figure 13. Same as Fig. 7 but for benzene (top panels), toluene (middle panels) and xylene (bottom panels).

Table 1. Summary of the statistical comparison between observed and simulated concentrations (ppt for aromatics, ppb for ozone). MMOD and MOBS represent the mean values for the SAPRC simulation and the observation, respectively. MRB is the relative bias of model results defined as: (MMOD – MOBS)/MOBS. SMOD and SOBS are their standard deviations. TCOR and SCOR are the temporal and spatial correlations between model results and measurements.

| Species | Network | Num of sites | Time resolution | MMOD (MRB) | MOBS | SMOD | SOBS | TCOR | SCOR |
|---------|---------|------|-----------------|------------|------|------|------|------|------|
| Benzene | CARIBIC | 1241 | Instantaneous | 12.3 (-23%) | 16.0 | 4.2 | 15.8 | - | 0.31 |
| | EEA | 22 | Annual mean | 131.6 (-32%) | 194.0 | 32.1 | 118.4 | - | 0.49 |
| | EMEP | 14 | Monthly | 106.5 (-36%) | 166.4 | 38.7 | 71.7 | 0.77 | 0.44 |
| | CALNEX | 7708 | Instantaneous | 66.1 (15%) | 57.7 | 78.3 | 57.7 | - | 0.51 |
| | KCMP | 1 | Hourly | 99.9 (9%) | 91.5 | 92.6 | 56.7 | 0.65 | - |
| Toluene | CARIBIC | 789 | Instantaneous | 1.5 (-58%) | 3.6 | 0.7 | 7.5 | - | 0.36 |
| | EEA | 6 | Annual mean | 180.9 (-25%) | 240.3 | 66.8 | 59.4 | - | 0.41 |
| | EMEP | 12 | Monthly | 113.2 (-15%) | 133.1 | 47.3 | 66.2 | 0.81 | 0.47 |
| | CALNEX | 7708 | Instantaneous | 80.6 (10%) | 73.2 | 179.7 | 131.9 | - | 0.46 |
| | KCMP | 1 | Hourly | 121.2 (114%) | 56.7 | 191.4 | 54.7 | 0.51 | - |
| Xylenes | EMEP | 8 | Monthly | 78.4 (85%) | 42.3 | 34.5 | 41.9 | 0.78 | 0.48 |
| $C_8$ aromatics | CALNEX | 7708 | Instantaneous | 28.8 (-41%) | 48.6 | 112.2 | 97.2 | - | 0.39 |
| | KCMP | 1 | Hourly | 88.9 (-2%) | 90.3 | 119.2 | 79.5 | 0.46 | - |
| Ozone | WDCGG | 64 | Monthly | 28.6 (-16%) | 34.1 | 12.8 | 14.2 | 0.68 | 0.54 |
| | EMEP | 130 | Monthly | 27.7 (-9%) | 30.6 | 13.2 | 10.3 | 0.76 | 0.52 |

Table 2. Annual and seasonal mean changes (%) in modeled surface as well as tropospheric concentrations from the Base to the SAPRC simulation. Also shown are the numbers for northern hemisphere (NH) and southern hemisphere (SH).

| Species | Annual | | MAM | | JJA | | SON | | DJF | |
|---|---|---|---|---|---|---|---|---|---|---|
| | Surface (NH, SH) | Trop (NH, SH) | Surface (NH, SH) | Trop (NH, SH) | Surface (NH, SH) | Trop (NH, SH) | Surface (NH, SH) | Trop (NH, SH) | Surface (NH, SH) | Trop (NH, SH) |
| NO | -0.2% (-0.2%, -1.4%) | 0.6% (0.8%, -0.2%) | -0.4% (-0.3%, -1.7%) | 0.7% (0.9%, -0.3%) | -1.3% (-1.3%, -1.2%) | -0.1% (-0.1%, -0.1%) | -1.5% (-1.5%, -1.3%) | -0.5% (-0.5%, -0.3%) | 0.8% (0.9%, -1.6%) | 1.6% (2.0%, -0.3%) |
| $O_3$ | 0.9% (1.2%, 0.3%) | 0.4% (0.6%, -0.1%) | 1.1% (1.6%, 0.3%) | 0.5% (0.8%, -0.1%) | 0.6% (0.9%, 0.2%) | 0.3% (0.5%, -0.1%) | 0.8% (1.1%, 0.4%) | 0.4% (0.6%, -0.1%) | 1.0% (1.3%, 0.3%) | 0.4% (0.6%, -0.1%) |
| CO | 0.8% (0.5%, 1.3%) | 1.0% (0.7%, 1.4%) | 0.5% (0.2%, 1.1%) | 0.7% (0.4%, 1.3%) | 1.1% (0.8%, 1.4%) | 1.2% (1.0%, 1.5%) | 1.1% (0.9%, 1.5%) | 1.3% (1.1%, 1.6%) | 0.5% (0.3%, 1.0%) | 0.7% (0.5%, 1.2%) |
| $HNO_3$ | 1.1% (1.3%, -0.6%) | 0.3% (0.7%, -0.9%) | 1.2% (1.3%, -0.4%) | 0.4% (0.7%, -0.9%) | 0.7% (0.9%, -0.6%) | -0.1% (0.2%, -1.0%) | 1.0% (1.4%, -0.7%) | 0.2% (0.7%, -1.0%) | 1.4% (1.6%, -0.7%) | 0.6% (1.1%, -0.8%) |
| $NO_2$ | 1.0% (1.0%, 0.2%) | 2.1% (2.4%, 0.7%) | 0.8% (0.8%, 0.3%) | 1.8% (2.0%, 0.8%) | -0.2% (-0.3%, 0.1%) | 0.6% (0.6%, 0.8%) | 0.5% (0.6%, 0.2%) | 1.3% (1.5%, 0.5%) | 2.0% (2.1%, 0.2%) | 3.6% (4.0%, 0.5%) |
| $NO_3$ | -0.9% (-0.6%, -2.7%) | -4.1% (-4.5%, -3.5%) | -1.5% (-1.3%, -2.7%) | -5.6% (-7.0%, -3.0%) | -0.9% (-0.5%, -2.5%) | -3.7% (-4.3%, -3.0%) | -0.5% (-0.1%, -2.6%) | -3.4% (-3.4%, -3.6%) | -0.8% (-0.5%, -3.6%) | -4.1% (-4.2%, -4.5%) |
| BENZ | -0.5% (-0.6%, 0.6%) | -0.4% (-0.6%, 1.4%) | -0.9% (-1.0%, 0.7%) | -1.0% (-1.1%, 1.7%) | 0.1% (-0.1%, 0.5%) | 0.7% (0.5%, 1.0%) | -0.1% (-0.2%, 0.8%) | 0.2% (-0.1%, 1.6%) | -0.6% (-0.6%, 0.9%) | -0.6% (-0.7%, 2.0%) |
| TOLU | -1.2% | -1.9% | -1.5% | -2.8% | -0.8% | -0.9% | -1.0% | -1.5% | -1.3% | -1.9% |

| | | | | | | | | | |
|---|---|---|---|---|---|---|---|---|---|
| | (-1.3%, 0.1%) | (-2.0%, 0.4%) | (-1.6%, 0.3%) | (-3.0%, 0.8%) | (-1.0%, -0.2%) | (-1.2%, -0.1%) | (-1.1%, 0.2%) | (-1.6%, 0.6%) | (-1.3%, 0.4%) | (-2.0%, 1.3%) |
| XYLE | -1.4% (-1.5%, -0.3%) | -2.3% (-2.3%, -0.2%) | -1.2% (-1.2%, -0.2%) | -2.1% (-2.2%, 0.3%) | -1.2% (-1.3%, -0.6%) | -1.5% (-1.6%, -0.9%) | -1.6% (-1.7%, -0.1%) | -2.3% (-2.4%, 0.2%) | -1.5% (-1.5%, -0.1%) | -2.4% (-2.4%, 0.5%) |
| OH | 1.1% (1.6%, 0.3%) | 0.2% (0.6%, -0.3%) | 1.4% (1.9%, 0.3%) | 0.4% (0.8%, -0.4%) | 1.2% (1.3%, 0.5%) | 0.3% (0.5%, -0.2%) | 0.9% (1.5%, 0.3%) | 0.1% (0.4%, -0.4%) | 1.0% (2.1%, 0.2%) | 0.1% (0.9%, -0.3%) |
| $HO_2$ | 3.0% (3.2%, 2.8%) | 1.3% (1.4%, 1.2%) | 2.9% (2.8%, 3.1%) | 1.4% (1.5%, 1.2%) | 3.3% (3.2%, 3.6%) | 1.3% (1.2%, 1.6%) | 3.1% (3.4%, 2.8%) | 1.3% (1.5%, 1.2%) | 2.8% (3.7%, 2.2%) | 1.2% (1.9%, 0.9%) |
| $OH/ HO_2$ | -1.4% (-1.0%, -1.7%) | -0.9% (-0.7%, -1.3%) | -1.2% (-1.1%, -1.9%) | -0.8% (-0.5%, -1.4%) | -1.6% (-1.1%, -2.0%) | -1.0% (-0.7%, -1.6%) | -1.4% (-0.9%, -1.9%) | -1.0% (-0.8%, -1.4%) | -1.1% (-0.5%, -2.1%) | -0.8% (-0.6%, -1.3%) |

1  Table 3. Annual and seasonal mean model ozone biases for the Base and the SAPRC case, compared to
2  measurements from WDCGG and EMEP.

| Species | Annual | | MAM | | JJA | | SON | | DJF | |
|---|---|---|---|---|---|---|---|---|---|---|
| (ppb) | Base | SAPRC | Base | SAPRC | Base | SAPRC | Base | SAPRC | Base | SAPRC |
| WDCGG | -6.0 | -5.4 | -9.0 | -8.4 | -0.4 | 0.1 | -2.5 | -2.1 | -11.9 | -11.5 |
| EMEP | -3.5 | -2.8 | -5.5 | -4.7 | 4.5 | 5.2 | 0.3 | 0.8 | -13.1 | -12.8 |

