# Peer review of "Global tropospheric effects of aromatic chemistry with the SAPRC-11"

_Geoscientific Model Development, 2018_

## Referee Comment (RC1) · Anonymous Referee #1 · 12 Sep 2018

**General Comments**

The manuscript by Yan et al. describes the implementation and impacts of an updated scheme for oxidation of aromatics (SAPRC-11) in the GEOS-Chem chemical transport model. The authors have provided a comprehensive overhaul to the previously very simplified benzene, toluene, and xylene chemistry. The updates are described in sufficient detail to allow reproducibility. The new simulation has been evaluated using both aircraft and surface observations and overall shows good agreement for aromatics and a reduction in model bias for ozone. The authors also quantify the impacts on related species including NOx, OH, and ozone, and show that there are small global impacts

but significant regional impacts (especially over anthropogenic source regions).

Overall, the updated chemistry is a valuable and important addition to a widely used global chemical transport model. The paper is generally well written, well structured, and easy to follow. The content and presentation are well suited to GMD, and I recommend publication once the following comments have been addressed.

**1. Model-observation comparisons should include the Base simulation**

Section 4 compares the SAPRC (updated) simulation to the observations and discusses differences and biases. However, the Base (original) simulation is never compared to the aromatics observations. There is a brief comparison to the ozone observations, although this is buried in Section 5.3. To clearly show the impacts of the new chemistry on the simulation, both the Base and SAPRC simulations should be compared to the observations in Section 4. The Base simulation should be added to Figures 2-5, and the discussion currently on Page 12 lines 29-39 should be moved to Section 4 (along with Table 3).

**2. SI tables should clearly identify new vs. updated species/reactions and should be consistent with GEOS-Chem nomenclature.**

Table S1 states it provides a list of "new model species", but several of these are existing species in GEOS-Chem. This should be clarified, and could be done by changing "new" to "relevant" in the table caption and adding a column for "New or Existing" to the table.

Similarly, Table S2 states it lists "new reactions and rate parameters", but again some reactions are currently in GEOS-Chem (presumably the rate parameters have been updated). Clarity is needed around what is new in the mechanism.

Finally, the species names in the SI do not match the GEOS-Chem conventions for existing species. Just a few examples of what GEOS-Chem uses: MO2 (not MEO2), CH2O (not HCHO), ALD2 (not CCHO), and many more. This work will be much more

usable by the GEOS-Chem community if the species list is updated to match. Existing species names are given at http://wiki.seas.harvard.edu/geos-chem/index.php/Species_in_GEOS-Chem

The SI tables should be updated where relevant to match.

**3. More details are needed to understand and be able to reproduce the model-observation comparisons**

Section 3 describes the observations used and, to some extent, the method in which the model was sampled for the comparisons. Some details are missing here that would be necessary for one to reproduce this work. Specifically, I had the following questions:

- CALNEX observations are at 1 second / 100 m resolution (pg 6, lines 20-21). This is much higher resolution than the model (2x2.5 degrees, timestep on order of minutes). Have the aircraft data then been averaged to the model resolution? If not the statistics will be biased by comparing multiple observation points to a single model grid point, especially as there will not be equal observation points in a given gridbox / timestep.

- For CARIBIC comparisons model output is sampled along the flight track (pg 6, lines 31-33). If this is the case, then why are model annual means used for the comparison? Shouldn't these be treated in the same way as CALNEX?

- Why are monthly means used for EMEP comparisons but annual means used for EEA comparisons (pg 7, lines 11-12 and 21)?

- Why are urban and suburban sites excluded from EEA comparisons (pg 7 lines 18-19) but not excluded from AQS comparisons (pg 10, line 2)?

- For KCMP, the paper specifies use of hourly observations (pg 7, line 26); are hourly model values also used?

- For KCMP sampling at 185m (pg 7, line 25), what box is the model sampled from, and how does that model layer compare to the 185m sampling height? I presume it wouldn't be the lowest model layer, since that is centred at 65m (pg 7 line 36).

- Why are so many more EMEP sites used for ozone (130) than for aromatics (8-14) (Table 1)?

- It would be useful to include a table providing sites and location information in the SI, especially since some stations have been excluded. This is probably not feasible for the large number of ozone sites, but would be for the aromatics data.

**4. Difference maps should be included for benzene, toluene, and xylene**

Although they are the focus of the paper, no maps of the aromatics spatial distribution are shown (except emissions), although they are hinted at on pg 11, lines 25-27. It seems to me critical to include figures analogous to Fig. 6 and 7 but for benzene, toluene, and xylene.

**5. Some comments are needed on the likely impact of changing aromatic emissions**

Anthropogenic aromatics emissions are from the Year 2000, while simulations and observations are for 2005 and 2010-2011. There are likely to have been significant changes in emissions and their distributions over the decade (briefly noted by authors on pg 8, lines 20-21). This is not a problem per se, but the paper would benefit from discussion of the likely changes and how they would benefit the results shown here (i.e. have aromatics gone up, in which case this work provides an upper limit? or the opposite?).

**Specific Comments**

Pg 2, lines 17-18: can some references be provided to back this up?

Pg 2, lines 29-30: is the overestimate global, or region-specific?

[Figure]

Pg 2, line 35: the introduction has jumped from models in general to GEOS-Chem specifically, so GEOS-Chem needs some introduction here

Pg 3, line 4: It would be better to use the updated GEOS-Chem versioning, which would make this version 12.0.0 rather than 11-02 (although technically the same).

Pg 3, line 9: I feel the equation would be easier to understand as aromatic + OH + NO = inert tracer (rather than "-NO" on the right-hand side)

Pg 3, lines 31+: what model time step is used?

Pg 4, lines 10-17: unclear why CO emissions are discussed here when CO is not a focus of the paper (and never shown later). If included here, would need to also include non-anthro CO sources (e.g. chemical production). Also, emissions from ships and aircraft missing. It might make sense to start this section with the NMVOC emissions rather than CO/NOx as they are the focus.

Pg 4, line 28: please specify species for the "aromatics" source – is this just benzene + toluene + xylene, or are other species included? Also "(71 Tg C)" can be deleted as it is given in the previous sentence (line 27).

Pg 5, line 15: "... which is consistent with the recent literature." More details are needed. What specifically does the SAPRC-11 mechanism reconcile that is/was missing from other mechanisms?

Pg 5, lines 21-33: I find this description hard to follow and hard to relate to what is in the tables in the SI. I think it would be helpful to give an example that traces the oxidation of one aromatic through these different production pathways.

Pg 6, lines 7-12: This is a little confusing and would suggest rephrasing. Is there a separate 6-month spin-up for each scenario (Base and SAPRC)? Is July-December 2009 also a spin-up period? For the sentence about initial conditions, does this mean that there is a $4°x5°$ spin-up run from Jan-Jun 2004 followed by a $2°x2.5°$ spin-up run for Jul-Dec 2004?

Pg 7, lines 13-14: The model speciation of xylenes should be clarified in the earlier section 2.2 about the mechanism.

Pg 8, lines 13-21: The model-observation difference would be a useful metric to include in Table 1.

Pg 8, line 30: Why are these 6 sites the ones used?

Pg 9, lines 13, 19: what are the lifetimes for benzene and toluene?

Pg 9, lines 23-34: any comment about the different profile shapes in the lower troposphere? What about the overly rapid benzene drop-off with altitude? Does that imply the modelled benzene lifetime is too short?

Pg 9, line 25: any comment on why winter shows an increase when the other seasons do not?

Pg 10, line 28: "The free tropospheric increases are largest in the remote northern regions" – I don't see this in Fig. 7. Instead it looks like the NO increases are about the same from 30S-90N.

Pg 10, lines 31-32: Rephrase this sentence as the start suggests it is about the surface NO2 but then it ends with "throughout the troposphere."

Pg 11, lines 1-3: Because of the different color scales, the overall NOx changes are not obvious in Figs 6 and 7. I'd suggest adding another panel to show the total NOx change.

Pg 11, lines 4-7: any comments on what is driving the NO3 global decreases and regional increases?

Pg 11, lines 8-11: Table 2 and the associated discussion in the text would be easier to follow if it were presented as a figure (e.g. a set of bar charts) rather than a densely packed table. Also, at the moment it includes species that are not discussed elsewhere in the text.

Pg 11, lines 31: Might be useful for this discussion to include the OH/HO2 ratio in the table (or figure)

Pg 12, lines 10-11: Please comment on why the ozone declines in biomass burning regions. Why have these changed in ways that are different from anthropogenic dominated regions? How can you tell that the changes are induced by biomass burning dominance rather than biogenic emissions dominance? If the former, I'm surprised not to see the same effects in boreal regions and in southern Africa.

Pg 12, lines 13-22: The reasons for the ozone increases are described, but what is causing the ozone decreases?

Pg 12, lines 27-28: Simulated production and loss rates could be used to test this.

Pg 12, lines 37-39: I think the conclusion here is that the halogen chemistry would bring the US ozone back down to the point that addition of aromatics would be a net improvement. If that's so, please make that point explicit. It also left me wondering what the impacts of the halogens would be outside of the US, where the biases shown in this work are already negative – would they become worse?

Pg 13, lines 6-9: It would be worth adding a panel to Figs. 8 and 9 to show the changes to the odd oxygen family. A panel for PAN would also be useful for the subsequent discussion.

Pg 13, line 25: which "organic nitrates" are referred to here? Is this PAN and analogues (PBZN)? Or does this refer to other organic nitrates like alkyl nitrates? It is not clear where in Table S2 one is meant to look for the chemistry of these nitrates.

Pg 13, lines 26-29: what NOx recycling is assumed in the model? Is this an effect that the authors have looked at (if so, can it be shown or described in more detail?), or does this refer to knowledge from existing literature (if so, references are needed...)?

Figure 2: Would be easier to interpret if common sites were aligned for the 3 species. (e.g. Zingst common between benzene and xylene, so move to upper left for xylene to

match location for benzene, etc.)

**Technical Comments**

Title: GMD requires specifying model version number in addition to name ("GEOS-Chem version 9-02")

Pg 5, line 5: suggest changing "true" to "the case"

Pg 5, line 6: change "v11-02" to "12.0.0" if changed above

Pg 5, line 10 (and elsewhere): the Carter and Heo (2013) reference is missing from the reference list

Pg 5, line 35: change "xylenols, phenols (XYNL)" to "xylenols and phenols (XYNL)" since XYNL represents both species.

Pg 6, line 8: suggest changing "based on the available observations" to "for comparison to the available observations"

Pg 6, line 18: suggest deleting "over the US" – this is too broad and already clear from the mention of California in the previous line.

Pg 7, line 7: change "though" to "through"

Pg 7, line 8: suggest changing "boundaries" to "geographic boundaries" (to clarify that this is not flux through e.g. air-land boundaries)

Pg 7, line 8: suggest changing to "locates measurement sites in locations where there are minimal..."

Pg 7, line 21: change "site" to "sites"

Pg 7, line 22: would be useful to add the location for the KCMP tall tower (e.g. US state?). Also does KCMP stand for something? Acronym is not defined.

Pg 8, line 2: suggest changing "part" to "section"

**[GMDD](GMDD)**
Pg 8, line 5: suggest changing "To do" to "For"

Pg 10, line 1: suggest deleting "relatively"

Pg 13, line 33: suggest changing "give" to "provide"

Figure 7: caption error; missing reference to NO3 and to middle plots

Figures 6-9: are these annual means? Which model year?

Table S1: Benzene, Toluene, and Xylene missing from species list

Table S2: What does "#" refer to?

---

## Short Comment (SC1) · 12 Sep 2018

Dear authors,

In my role as Executive editor of GMD, I would like to bring to your attention our Editorial version 1.1:

http://www.geosci-model-dev.net/8/3487/2015/gmd-8-3487-2015.html

This highlights some requirements of papers published in GMD, which is also available on the GMD website in the 'Manuscript Types' section:

http://www.geoscientific-model-development.net/submission/manuscript_types.html

[Figure]

In particular, please note that for your paper, the following requirements have not been met in the Discussions paper:

- "The main paper must give the model name and version number (or other unique identifier) in the title."

- "All papers must include a section, at the end of the paper, entitled 'Code availability'. Here, either instructions for obtaining the code, or the reasons why the code is not available should be clearly stated. It is preferred for the code to be uploaded as a supplement or to be made available at a data repository with an associated DOI (digital object identifier) for the exact model version described in the paper. Alternatively, for established models, there may be an existing means of accessing the code through a particular system. In this case, there must exist a means of permanently accessing the precise model version described in the paper. In some cases, authors may prefer to put models on their own website, or to act as a point of contact for obtaining the code. Given the impermanence of websites and email addresses, this is not encouraged, and authors should consider improving the availability with a more permanent arrangement. After the paper is accepted the model archive should be updated to include a link to the GMD paper."

Please include the version number of GEOS-Chem in the title of the revised manuscript. Additionally, please include information how to optain the GEOS-Chem Code into the Code Availability Section. Note, that it is not sufficient to only state that the code is available from author without stating reasons, why publication is not possible.

Yours,

Astrid Kerkweg

****************************************** ! models: name + v

---

## Referee Comment (RC2) · Anonymous Referee #2 · 26 Sep 2018

This paper reported an excellent timely effort updating aromatic VOC chemistry in GEOS-Chem, a widely used global chemistry model. The effort is very useful for the community given the importance of aromatics in regional and global chemistry and the potential limitation of the existing chemical mechanism included in GEOS-Chem. The paper describes the motivation, methodology in a very clear fashion. The key model results (e.g., NOx, HOx, ozone) are selected appropriately and discussed thoroughly, and are interpreted carefully by recognizing both the strengths and the potential limitations of the model setup and input data. A very comprehensive model evaluation has been carried out using data from multiple global and regional networks/programs. I recommend publication after my following comments are considered.

Major comments

- The use of AQS ozone data in model evaluation is inappropriate and should be removed

It is simply inappropriate to directly compare urban and suburban AQS ozone observations near the surface ($\sim$ 10 m) to GEOS-Chem ozone at 65 m height with 2x2.5 deg horizontal resolution. The model evaluation results using AQS data is not only meaningless but also misleading, especially when these results are discussed along with other networks in remote environments, where the model evaluation is actually appropriate and meaningful. Thus, I strongly suggest the authors remove the model evaluation with AQS ozone and focus on using networks over rural and clean environments.

- The adoption of SAPRC-11 and uncertainties in knowledge of aromatic chemistry

The paper describes the SAPRC-11 mechanism itself in detail and the method to include it into GEOS-Chem clearly. However, it is yet to be more clear why it is chosen instead of other options, such as the condensed MCM mechanism. One thing about SAPRC is the use of maximum ozone formation as a primary metric in the chamber experiment benchmark, and the mechanism has been primarily used and evaluated in regional CTMs such as CMAQ and CAMx, at much finer resolution (i.e., a few kilometers). I think the present paper is the first to use it in a global model. Therefore, the authors should have some words justifying the approach. Also, are there other considerations behind the simplified GEOS-Chem aromatic chemistry, in addition to minimizing the number of reactions? Moreover, it should be noted that our knowledge about the very complex aromatic chemistry itself is not complete. For instance, how would the uncertainties in the yields of di-carbonyls and radical recycling affect the mechanism and the model simulations? The simplified chemistry in GEOS-Chem does not have radical cycling, but are there any assumptions/uncertainties in SAPRC-11 about radical cycling that might have impact on the results too?

Adding some discussions on these above questions would make the paper even stronger.

Minor comments

P2, L19-L21: "Despite the potentially important influence of aromatic compounds on global atmospheric chemistry, their effect on tropospheric ozone formation in polluted urban areas remains largely unknown." "Unknown" is an overstatement of the issue to me. Aromatic VOCs have long been recognized as a key player in urban photochemistry, forming PAN and ozone, and SOA, despite the uncertainties with the chemistry (and emissions).

P2, L21-L22: "The main source and sink processes of tropospheric ozone are photochemical production and loss, respectively (Yan et al., 2016)" Other references such as textbook by Seinfeld and Pandis (2006) would be more appropriate in this sentence.

P2, L33: ". . . including the parameterization of small-scale processes and their feedbacks to global-scale chemistry (Yan et al., 2014; Yan et al., 2016)." Other references should be added in addition to these two.

P5, L27: "The OH-aromatic adduct is reaction with O2. . ." This sentence needs rephrase.

P6, L13: Have the authors considered evaluating species other than ozone and aromatics, such as aircraft measurements of HOx (CalNex probably has some HOx measurements)?

P7, L32: Data download link does not work (last access 9/26/18) http://aqsdr1.epa.gov/aqsweb/aqstmp/airdata/download_files.html

P7, L36: see my first major comment.

P12, L30: The discussions at AQS sites should be removed.

P13, Section 5.4: See my second major comment. I suggest adding discussions of uncertainty in knowledge of aromatic chemistry and the considerations and assumptions in SAPRC-11.

Table 2: I suggest add numbers for NH and SH

[Figure]

---

## Referee Comment (RC3) · Anonymous Referee #3 · 19 Oct 2018

This paper describes the implementation of the (State-wide Air Pollution Research Center) SAPRC-11 representation of BTEX mono-aromatic chemistry into the 9-02 version of the GEOS-Chem global chemical transport model. This is timely, given the importance of aromatic chemistry in the global atmosphere, with respect to air quality (i.e. ozone and other secondary photochemical pollutants) and secondary organic aerosol formation. Model evaluations have been carried out against a significant, wide ranging observational database (both long term ground and aircraft flight path measurements) of aromatics and ozone concentrations. Model analysis of the effects of the new chemistry on the important model outputs of O3, NOx and HOx have been carried out and discussed with respect to global and regional biases.

Overall, this paper is reasonably well written (although lacking in some detail, especially with respect to the specific aromatic chemistry implemented – see discussion) and will be useful to the global CTM community. It is in good scope for GMD. I recommend publication after the following comments have been addressed.

**(1)      More detailed description of aromatic photochemistry implemented (base case and updated aromatic chemistry).**

It would be useful to the reader to have a more detailed description of the aromatic chemistry represented in the Base model as well as the SAPRC update. For example, a simplified schematic showing the structure of the different mono-aromatics and how reaction with OH leads to initial OH-adducts (and OH abstraction products from OH attack at the methyl groups) that can then convert to different ring retaining and ring opening products, though the representative RO2 species formed from subsequent reactions with O2 and NO, leading to significant O3 production. This chemistry is briefly discussed in the text, and in a way that is only understandable from an experienced GEOS-Chem user (form the base case at least) but should be given in more detail as this important chemistry is the subject of this paper.

Also, when discussing the SAPRC aromatic-ozone chemistry in Section 5.4, it would be useful to provide the basic photochemical ozone formation chemistry equations (including PAN formation) so that the discussion in the text can be followed more closely.

**(2)      Discussion of uncertainties in the aromatic chemistry and comparisons with other, more detailed mechanisms.**

There is little discussion about the development of the SAPRC chemical mechanisms, the uncertainties in the specific aromatic chemistry implemented and how the chemistry compares to other widely used detailed chemical schemes.

SAPRC was originally developed in order to model one day photochemical smog episodes typical of, for example, Los Angeles and other North American urban centres. SAPRC is a highly efficient and compact chemical mechanism, therefore can be implementation into CTMs, but is based on lumped chemistry, which is partly optimised on empirical fitting to smog chamber experiments that are representative to US one day conditions. Therefore, some discussion should be made with respect to applications of this optimised chemistry

outside these optimisation conditions – e.g. SH tropics. How does the SAPRC chemistry compare to more detailed chemical mechanisms, which are based upon more fundamental laboratory and theoretical data, which are used for policy and scientific modelling multi-day photochemical ozone formation that is experienced over Europe – e.g. the Master Chemical Mechanism?

It is also clear from the literature and atmospheric chamber model-mechanism comparisons that aromatic chemistry is still far from being completely understood. For example, Bloss et al., (2005) show that for alkyl substituted mono-aromatics, comparisons to chamber experiment over a range of VOC/NOx conditions that the chemistry under predictions the reactivity of the system but over predicts the amount of O3 produced (model shows more NO to NO2 conversion than on the experiments). How does the uncertainties in the fundamental aromatic chemistry effect the modelling shown here?

**(3)      Specific Comments**

References are not in alphabetical order

How much more computational effort does implementing SAPRC-11 chemistry add in terms of model simulation times?

Introduction – better referencing of the aromatic literature needed, e.g. Atkinson and Arey (2003) and Calvert et al., (2002).

*"Despite the potentially important influence of aromatic compounds on global atmospheric chemistry, their effect on tropospheric ozone formation in polluted urban areas remains largely unknown"*. This statement is simply not true. There is a large amount of literature on this subject and original policy based emission reactivity indexes such as MIR (which is based on SAPRC) and POCP (which is based on MCM) show the importance of aromatic chemistry to ozone formation in the US and Europe respectively.

*"Current global CTMs reproduce much of the observed regional and seasonal variability in tropospheric ozone concentrations."* This is a broad statement and needs to be qualified. Surely the very reason that you are carrying out this study is that this is not true?!

*"GEOS-Chem"* needs to be defined in more detail. References to v9-02 and v11-02 need to be added.

*"SAPRC-11"* also needs better defining

2.2. Updated aromatic chemistry – *"Moreover,SAPRC-11 is able to reproduce the ozone formation from aromatic oxidation that is observed in environmental chamber experiments"*. Under what conditions? (VOC/NOx)

3.2 Aromatic Surface Measurements – where is the KCMP tower? Define.

5.1 NOy Species – *"Combing the changes in NO…"* ???

5.2 OH and HO2 – "*Compared to the Base simulation, OH increases slightly by 1.1% at the surface in the SAPRC simulation (Fig. 8 and Table 2)."* Discussion of the observed deceases?

"*In these locations, the peroxy radicals formed by aromatic oxidation react with NO2 and HO2*" – surely NO and HO2?

"*This in turn influences OH, as the largest photochemical sources of OH are the photolysis of O3 as well as the reaction of NO with HO2*" – largest photochemical sources of OH *in the model.*

"*Seasonally, a few surface locations see OH concentration increases of more than 10% during April–August (not shown), including parts of the eastern US, central Europe, eastern Asia and Japan."* There seem to be a few points in the text where interesting model results are eluded to but "not shown". Could some of these not be included in the supplementary?

5.3 Ozone – "*The aromatics transported to the upper troposphere may cause net consumption of tropospheric OH and NOx, which can further reduce ozone production".* How?

Could other atmospherically important species that are in aromatic chemistry be compared to the observations – specifically the detailed data sets from CALNEX – e.g. HOx, HCHO, PAN, Glyoxal and Methyl Glyoxal? These are all important tracers of active photochemistry.

References

Atkinson, R. and Arey, J.: Atmospheric degradation of volatile organic compounds, Chem. Rev., 103, 4605–4638, https://doi.org/10.1021/cr0206420, 2003.

Bloss, C., Wagner, V., Bonzanini, A., Jenkin, M. E., Wirtz, K., Martin-Reviejo, M., and Pilling, M. J. Evaluation of detailed aromatic mechanisms (MCMv3 and MCMv3.1) against environmental chamber data, Atmos. Chem. Phys., 5, 623-639, 2005.

Calvert, J. G., Atkinson, R., Becker, K. H., Kamens, R. M., Seinfeld, J. H., Wallington, T. J., and Yarwood, G.: The mechanisms of atmospheric oxidation of aromatic hydrocarbons, Oxford University Press, New York, 2002.

---

## Author Comment (AC1) · 10 Nov 2018

Anonymous Referee #1 General Comments The manuscript by Yan et al. describes the implementation and impacts of an updated scheme for oxidation of aromatics (SAPRC-11) in the GEOS-Chem chemical transport model. The authors have provided a comprehensive overhaul to the previously very simplified benzene, toluene, and xylene chemistry. The updates are described in sufficient detail to allow reproducibility. The new simulation has been evaluated using both aircraft and surface observations and overall shows good agreement for aromatics and a reduction in model bias for ozone. The authors also quantify the impacts on related species including NOx, OH,

and ozone, and show that there are small global impacts but significant regional impacts (especially over anthropogenic source regions). Overall, the updated chemistry is a valuable and important addition to a widely used global chemical transport model. The paper is generally well written, well structured, and easy to follow. The content and presentation are well suited to GMD, and I recommend publication once the following comments have been addressed. We thank the reviewer for comments, which have been incorporated to improve the manuscript. 1. Model-observation comparisons should include the Base simulation Section 4 compares the SAPRC (updated) simulation to the observations and discusses differences and biases. However, the Base (original) simulation is never compared to the aromatics observations. There is a brief comparison to the ozone observations, although this is buried in Section 5.3. To clearly show the impacts of the new chemistry on the simulation, both the Base and SAPRC simulations should be compared to the observations in Section 4. The Base simulation should be added to Figures 2-5, and the discussion currently on Page 12 lines 29-39 should be moved to Section 4 (along with Table 3). Thanks for the comment from referee. We have added the modeled spatial distributions of annual mean surface (revised Figure 12) and zonal average latitude-altitude distributions of annual mean (revised Figure 13) benzene, toluene, and xylene simulated in the Base case for the year 2005. Also shown in these figures are the respective changes from Base to SAPRC. These two figures show that the changes from Base to SAPRC in annual average surface aromatics and zonal average aromatics are less than 2% for individual species. The differences between Base and SAPRC is much smaller than the modeled bias in SAPRC compared to aromatics observations. Thus we have kept the ozone comparison with Base and SAPRC in Sect. 5.3 to show the effects from SAPRC on ozone simulation. 2. SI tables should clearly identify new vs. updated species/reactions and should be consistent with GEOS-Chem nomenclature. Table S1 states it provides a list of "new model species", but several of these are existing species in GEOS-Chem. This should be clarified, and could be done by changing "new" to "relevant" in the table caption and adding a column for "New or Existing" to the

table. In the revised Table S1, we have identified new vs. existing species by changing "new" to "relevant" in the table caption and adding a column for "New or Existing" to the table. Similarly, Table S2 states it lists "new reactions and rate parameters", but again some reactions are currently in GEOS-Chem (presumably the rate parameters have been updated). Clarity is needed around what is new in the mechanism. In the revised Table S2, we have identified new vs. updated reactions by changing "new" to "relevant" in the table caption and adding a column for "New or Updated" to the table. The updated reaction is meant to update the rate parameters. Finally, the species names in the SI do not match the GEOS-Chem conventions for existing species. Just a few examples of what GEOS-Chem uses: MO2 (not MEO2), CH2O (not HCHO), ALD2 (not CCHO), and many more. This work will be much more usable by the GEOS-Chem community if the species list is updated to match. Existing species names are given at http://wiki.seas.harvard.edu/geos-chem/index.php/ Species_in_GEOS-Chem. The SI tables should be updated where relevant to match. We have updated the species list in Table S1 and Table S2 to match the GEOS-Chem conventions for existing species. 3. More details are needed to understand and be able to reproduce the model observation comparisons Section 3 describes the observations used and, to some extent, the method in which the model was sampled for the comparisons. Some details are missing here that would be necessary for one to reproduce this work. Specifically, I had the following questions: • CALNEX observations are at 1 second / 100 m resolution (pg 6, lines 20-21). This is much higher resolution than the model (2x2.5 degrees, timestep on order of minutes). Have the aircraft data then been averaged to the model resolution? If not the statistics will be biased by comparing multiple observation points to a single model grid point, especially as there will not be equal observation points in a given gridbox / timestep. We have added the information in the revised Sect. 3.1: "For comparison to the model results, we averaged the high temporal-spatial resolution observations to the model resolution." • For CARIBIC comparisons model output is sampled along the flight track (pg 6, lines 31-33). If this is the case, then why are model annual means used for the

comparison? Shouldn't these be treated in the same way as CALNEX? We first averaged the measurements to the model output resolution. Then in comparison, we use annual means of observations and model data along the flight track. In the revised description, we have added the details: "To evaluate our results, measurements are averaged to the model output resolution. Vertically, results from GEOS-Chem model simulations at the 250 hPa level are used to compare with observations between 200–300 hPa. Then the annual means of observations and model data sampled along the flight tracks are used in the comparison." • Why are monthly means used for EMEP comparisons but annual means used for EEA comparisons (pg 7, lines 11-12 and 21)? We used monthly means for EMEP comparisons but annual means for EEA comparisons, mainly because that the EEA measurements have much more missing data than the EMEP observations. • Why are urban and suburban sites excluded from EEA comparisons (pg 7 lines 18-19) but not excluded from AQS comparisons (pg 10, line 2)? Based on the comment from referee#2, in the revised text, we have removed the model evaluation with AQS ozone measurements, because that it is inappropriate to directly compare AQS ozone observations near the surface (âĹij 10 m) to GEOS-Chem ozone at 65 m height with 2x2.5 deg horizontal resolution. • For KCMP, the paper specifies use of hourly observations (pg 7, line 26); are hourly model values also used? We averaged the hourly observations to monthly values and then compared to the monthly model results. We have added the information in the revised sentence: "We averaged the hourly observations of benzene, toluene and C8 (xylenes + ethylbenzene; here consistent with the model speciation) aromatics to monthly values and then used for our model evaluation." • For KCMP sampling at 185m (pg 7, line 25), what box is the model sampled from, and how does that model layer compare to the 185m sampling height? I presume it wouldn't be the lowest model layer, since that is centred at 65m (pg 7 line 36). We have added the information at the end of this paragraph: "Monthly mean simulations at the 990 hPa level (∼190 m) are used for comparison." • Why are so many more EMEP sites used for ozone (130) than for aromatics (8-14) (Table 1)? It is because that aromatics

downloaded from EMEP (http://www.nilu.no/projects/ccc/emepdata.html) are much less than ozone measurements. • It would be useful to include a table providing sites and location information in the SI, especially since some stations have been excluded. This is probably not feasible for the large number of ozone sites, but would be for the aromatics data. The sites and location information of aromatics data used here are described in detail in Cabrera-Perez et al., 2016 who download the raw data and provide the collated data. We have added this information in the revised text: "Further details of the sites and location information of EEA (and EMEP) used here are described in Cabrera-Perez et al., 2016." 4. Difference maps should be included for benzene, toluene, and xylene Although they are the focus of the paper, no maps of the aromatics spatial distribution are shown (except emissions), although they are hinted at on pg 11, lines 25-27. It seems to me critical to include figures analogous to Fig. 6 and 7 but for benzene, toluene, and xylene. We have added the modeled spatial distributions of annual mean surface (revised Figure 12) and zonal average latitude-altitude distributions of annual mean (revised Figure 13) benzene, toluene, and xylene simulated in the Base case for the year 2005. Also shown in these figures are the respective changes from Base to SAPRC. 5. Some comments are needed on the likely impact of changing aromatic emissions Anthropogenic aromatics emissions are from the Year 2000, while simulations and observations are for 2005 and 2010-2011. There are likely to have been significant changes in emissions and their distributions over the decade (briefly noted by authors on pg 8, lines 20-21). This is not a problem per se, but the paper would benefit from discussion of the likely changes and how they would benefit the results shown here (i.e. have aromatics gone up, in which case this work provides an upper limit? or the opposite?). Thanks for the comment from referee. We have added discussion in the revised Sect. 4.1: "Anthropogenic aromatics emissions are reported to have significant changes in emissions and their distributions over the decade by EDGARv4.3.2 (Crippa et al., 2018; http://eccad.aeris-data.fr/#DatasetPlace:EDGARv4.3.2$DOI). It shows that the total aromatics emission from anthropogenic source are enhanced by 5% (2005) and

14% (2011) compared to the year 2000. The model bias would be partly benefit from this emission increase with enhanced modeled mixing ratios of benzene and toluene." Specific Comments Pg 2, lines 17-18: can some references be provided to back this up? We have added the references of Lewis et al., 2013 and Cabrera-Perez et al., 2016. Pg 2, lines 29-30: is the overestimate global, or region-specific? We have revised this sentence: "However, some systematic biases can occur, most commonly an overestimation over the northern hemisphere" Pg 2, line 35: the introduction has jumped from models in general to GEOS-Chem specifically, so GEOS-Chem needs some introduction here We have revised this sentence: "Another motivation for the modeling comes from recent updates in halogen (bromine-chlorine) chemistry, which when implemented in GEOS-Chem, a global chemical transport model being used extensively for tropospheric chemistry and transport studies (Zhang and Wang, 2016; Yan et al., 2014; Shen et al., 2015; Lin et al., 2016), decrease the global burden of ozone significantly (by 14%; 2–10 ppb in the troposphere) (Schmidt et al., 2017)." Pg 3, line 4: It would be better to use the updated GEOS-Chem versioning, which would make this version 12.0.0 rather than 11-02 (although technically the same). We have modified the version. Pg 3, line 9: I feel the equation would be easier to understand as aromatic + OH + NO = inert tracer (rather than "-NO" on the right-hand side) We have modified the equation as aromatic + OH + NO = inert tracer Pg 3, lines 31+: what model time step is used? We have added the time step information: "The chemistry time step is 0.5 h, while the transport time step is 15 min in the model." Pg 4, lines 10-17: unclear why CO emissions are discussed here when CO is not a focus of the paper (and never shown later). If included here, would need to also include non-anthro CO sources (e.g. chemical production). Also, emissions from ships and aircraft missing. It might make sense to start this section with the NMVOC emissions rather than CO/NOx as they are the focus. We have removed the CO emission description and moved the NOx emission behind the NMVOC emission description. Pg 4, line 28: please specify species for the "aromatics" source – is this just benzene + toluene + xylene, or are other species included? Also "(71 Tg C)" can be deleted as it is given

in the previous sentence (line 27). We have added the species for the "aromatics" source and removed the "(71 Tg C)" in this sentence: "On a carbon basis, the global aromatics (benzene + toluene + xylenes) source accounts for $\sim$ 23% (16 TgC) of the total anthropogenic NMVOCs." Pg 5, line 15: ". . . which is consistent with the recent literature." More details are needed. What specifically does the SAPRC-11 mechanism reconcile that is/was missing from other mechanisms? We have added some details of major updates in the SAPRC-11: "SAPRC-11 is an updated version of the SAPRC-07 mechanism (Carter and Heo, 2013) to give better simulations of recent environmental chamber experiments. The SAPRC-07 mechanism underpredicted NO oxidation and O3 formation rates observed in recent aromatic-NOx environmental chamber experiments. The new aromatics mechanism, designated SAPRC-11, is able to reproduce the ozone formation from aromatic oxidation that is observed in environmental chamber experiments (Carter and Heo, 2013)." Pg 5, lines 21-33: I find this description hard to follow and hard to relate to what is in the tables in the SI. I think it would be helpful to give an example that traces the oxidation of one aromatic through these different production pathways. We have modified the description by taking toluene as an example: "In SAPRC-11, taking toluene as an example in Table S2, the reactions following abstraction lead to three different formation products: an aromatic aldehyde (represented as the BALD species in the model), a ketone (PROD2), and an aldehyde (RCHO). The largest yield of toluene oxidation is the reaction after OH addition of aromatic rings. The OH-aromatic adduct is reaction with O2 either forming HO2 and a phenolic compound (further consumed by reactions with OH and NO3 radicals), or to form an OH-aromatic-O2 adduct. The OH-aromatic-O2 adduct further undergos two competing unimolecular reactions to ultimately form OH, HO2, an $\alpha$-dicarbonyl (such as glyoxal (GLY), methylglyoxal (MGLY) or biacetyl (BACL)), a monounsaturated dicarbonyl co-product (AFG1, AFG2, the photoreactive products) and a di-unsaturated dicarbonyl product (AFG3, the non-photoreactive products) (Calvert et al., 2002)." Pg 6, lines 7-12: This is a little confusing and would suggest rephrasing. Is there a separate 6-month spin-up for each scenario (Base and SAPRC)? Is July-December

2009 also a spin-up period? For the sentence about initial conditions, does this mean that there is a 4◦x5◦ spin-up run from Jan-Jun 2004 followed by a 2◦x2.5◦ spin-up run for Jul-Dec 2004? We have revised these sentences: "Both simulations (Base and SAPRC) at 2.5° long. 2° lat. are conducted from July 2004 to December 2005, allowing for a 6-month spin-up for our focused analysis over the year of 2005 based on the available observations (Sect. 3). Initial conditions of chemicals are regridded from a simulation at 5° long. 4° lat. started from 2004 with another spin-up run from January to June 2004. For comparison with aromatics observations over the US in 2010–2011 (Sect. 3), we extend the simulations from July 2009 to December 2011 with July-December 2009 as the spin-up period." Pg 7, lines 13-14: The model speciation of xylenes should be clarified in the earlier section 2.2 about the mechanism. We have added the model speciation of xylenes in revised Sect. 2.1: "Here the model speciation of xylenes includes m-xylene, p-xylene, o-xylene and ethylbenzene (Hu et al., 2015)." Pg 8, lines 13-21: The model-observation difference would be a useful metric to include in Table 1. We have added the calculation of model-observation difference in revised Table 1. It is the MRB (relative bias of model results) defined as: (MMOD – MOBS)/MOBS. Pg 8, line 30: Why are these 6 sites the ones used? We have added the reason: "The sites are chosen as the first six stations with largest amount of data." Pg 9, lines 13, 19: what are the lifetimes for benzene and toluene? The lifetime of benzene is between several hours and several days, and toluene is between several days and several weeks irrespective of the time of year (http://www.nzdl.org/gsdlmod?a=p&p=home&l=en&w=utf-8). We have added the lifetime for toluene in the revised Sect. 4.2. Pg 9, lines 23-34: any comment about the different profile shapes in the lower troposphere? What about the overly rapid benzene drop-off with altitude? Does that imply the modelled benzene lifetime is too short? Thanks for this comment from referee. We have added the discussion in the revised Sect. 4.2: "The different profile shapes in the lower troposphere for benzene, toluene and C8 aromatics are mainly due to their different emissions and lifetime. The modeled overly rapid aromatics drop-off with altitude probably implies

the modelled aromatics lifetime is short." Pg 10, line 25: any comment on why winter shows an increase when the other seasons do not? We have added the discussion in the revised Sect. 5.1: "This winter increase versus decline in other seasons is probably attributed to the weaken photochemical reactions involving NOx in winter." Pg 10, line 28: "The free tropospheric increases are largest in the remote northern regions" – I don't see this in Fig. 7. Instead it looks like the NO increases are about the same from 30S-90N. We have revised this sentence as: "The free tropospheric NO increases are about the same from 30S-90N". Pg 10, lines 31-32: Rephrase this sentence as the start suggests it is about the surface NO2 but then it ends with "throughout the troposphere." We have revised this sentence as: "Figure 6 shows that simulated surface NO2 mixing ratios in the SAPRC scenario are enhanced over most locations across the globe". Pg 11, lines 1-3: Because of the different color scales, the overall NOx changes are not obvious in Figs 6 and 7. I'd suggest adding another panel to show the total NOx change. We have added the modeled spatial distributions of annual mean surface NOx (revised Figure 7) and zonal average latitude-altitude distributions of annual mean (revised Figure 8) NOx simulated in the Base case for the year 2005. Also shown in these figures are the respective changes from Base to SAPRC. Pg 11, lines 4-7: any comments on what is driving the NO3 global decreases and regional increases? We have added the discussion following this sentence: "The NO3 global decreases are mainly due to the consumption of NO3 by reaction with the aromatic oxidation products. However, the NO3 regional increases are probably caused by the enhanced regional atmospheric oxidation capacity." Pg 11, lines 8-11: Table 2 and the associated discussion in the text would be easier to follow if it were presented as a figure (e.g. a set of bar charts) rather than a densely packed table. Also, at the moment it includes species that are not discussed elsewhere in the text. Thanks for this comment from referee. We have kept the table in the revised manuscript, mainly because that the amount of data in Table 2 is large to be difficult presented as a figure and be also difficult to show the specific value in the bar charts. In the revised Table 2, we have removed the calculation results of species (H2O2 and

N2O5) that are not discussed in the text. Pg 11, lines 31: Might be useful for this discussion to include the OH/HO2 ratio in the table (or figure) We have included the OH/HO2 ratio in the revised Table 2. Pg 12, lines 10-11: Please comment on why the ozone declines in biomass burning regions. Why have these changed in ways that are different from anthropogenic dominated regions? How can you tell that the changes are induced by biomass burning dominance rather than biogenic emissions dominance? If the former, I'm surprised not to see the same effects in boreal regions and in southern Africa. Based on the recent data analysis, we cannot yet comment on why the ozone declines in regions dominated by biomass burning or biogenic emissions. We have revised this sentence to include the specific regions: "ozone declines in regions of South America, Central Africa, Australia and Indonesia over the tropics (30°SïĂ■30°N)." The reasons for the ozone decline are discussed below: "These decreases are probably related to the upward transport of aromatics by tropical convection processes. The aromatics transported to the upper troposphere may cause net consumption of tropospheric OH and NOx, which can further reduce ozone production." Pg 12, lines 13-22: The reasons for the ozone increases are described, but what is causing the ozone decreases? The reasons for the ozone decline are discussed below: "These decreases are probably related to the upward transport of aromatics by tropical convection processes. The aromatics transported to the upper troposphere may cause net consumption of tropospheric OH and NOx, which can further reduce ozone production." Pg 12, lines 27-28: Simulated production and loss rates could be used to test this. Thanks for this comment from referee. Regretfully, we did not output the modeled results of production and loss rates. Pg 12, lines 37-39: I think the conclusion here is that the halogen chemistry would bring the US ozone back down to the point that addition of aromatics would be a net improvement. If that's so, please make that point explicit. It also left me wondering what the impacts of the halogens would be outside of the US, where the biases shown in this work are already negative – would they become worse? Based on the comment from referee#2, in the revised manuscript, we have removed the model evaluation with

AQS ozone measurements and the discussion of halogen chemistry, because that it is inappropriate to directly compare urban and suburban AQS ozone observations near the surface (âĹij 10 m) to GEOS-Chem ozone at 65 m height with 2x2.5 deg horizontal resolution. Pg 13, lines 6-9: It would be worth adding a panel to Figs. 8 and 9 to show the changes to the odd oxygen family. A panel for PAN would also be useful for the subsequent discussion. We have added the modeled spatial distributions of annual mean surface PAN (revised Figure 7) and zonal average latitude-altitude distributions of annual mean PAN (revised Figure 8) simulated in the Base case for the year 2005. Also shown in these figures are the respective changes from Base to SAPRC. For the odd oxygen family (Ox), they are shown in revised Figure 10 and Figure 11. Pg 13, line 25: which "organic nitrates" are referred to here? Is this PAN and analogues (PBZN)? Or does this refer to other organic nitrates like alkyl nitrates? It is not clear where in Table S2 one is meant to look for the chemistry of these nitrates. We have added the specific species and the referred reactions shown in Table S2 in the revised sentence: "In addition, production of organic nitrates (PBZN (reactions of BR30 and BR31 in Table S2) and RNO3 (PO36)) in the model with SAPRC aromatics chemistry". Pg 13, lines 26-29: what NOx recycling is assumed in the model? Is this an effect that the authors have looked at (if so, can it be shown or described in more detail?), or does this refer to knowledge from existing literature (if so, references are needed. . .)? We have changed the "recycling of NOx" to "such re-release of NOx" in the revised sentence. The re-release of NOx process have described in the former sentence: "In addition, production of organic nitrates (PBZN (reactions of BR30 and BR31 in Table S2) and RNO3 (PO36)) in the model with SAPRC aromatics chemistry may also explain the increase in ambient NOx in the remote regions, due to the re-release of NOx from organic nitrates (as opposed to removal by deposition)." Figure 2: Would be easier to interpret if common sites were aligned for the 3 species. (e.g. Zingst common between benzene and xylene, so move to upper left for xylene to match location for benzene, etc.) We have moved the common sites to be aligned for the three species in the revised Figure 2. Technical Comments Title: GMD requires

specifying model version number in addition to name ("GEOSChem version 9-02") We have added the model version into the title. Pg 5, line 5: suggest changing "true" to "the case" Have changed. Pg 5, line 6: change "v11-02" to "12.0.0" if changed above Have changed. Pg 5, line 10 (and elsewhere): the Carter and Heo (2013) reference is missing from the reference list We have added the Carter and Heo (2013) reference in the reference list. Pg 5, line 35: change "xylenols, phenols (XYNL)" to "xylenols and phenols (XYNL)" since XYNL represents both species. Have changed. Pg 6, line 8: suggest changing "based on the available observations" to "for comparison to the available observations" Have changed. Pg 6, line 18: suggest deleting "over the US" – this is too broad and already clear from the mention of California in the previous line. Have deleted "over the US". Pg 7, line 7: change "though" to "through" Have changed. Pg 7, line 8: suggest changing "boundaries" to "geographic boundaries" (to clarify that this is not flux through e.g. air-land boundaries) Thanks for comment from referee. We have changed. Pg 7, line 8: suggest changing to "locates measurement sites in locations where there are minimal. . ." Have changed. Pg 7, line 21: change "site" to "sites" Have changed. Pg 7, line 22: would be useful to add the location for the KCMP tall tower (e.g. US state?). Also does KCMP stand for something? Acronym is not defined. We have added the location in the revised sentence: "The KCMP tall tower measurements (at 44.69°N, 93.07°W, Minnesota, US) have been widely used for studies". The the KCMP is the current Minnesota Public Radio. Pg 8, line 2: suggest changing "part" to "section" Have changed. Pg 8, line 5: suggest changing "To do" to "For" Have changed. Pg 10, line 1: suggest deleting "relatively" Have deleted. Pg 13, line 33: suggest changing "give" to "provide" Have changed. Figure 7: caption error; missing reference to NO3 and to middle plots We have added the reference to NO3 and to middle plots in caption of Figure 7. Figures 6-9: are these annual means? Which model year? Yes, they are annual means and for the year 2005. We have added the information in the captions. Table S1: Benzene, Toluene, and Xylene missing from species list We have added these three species in the revised Table S1. Table S2: What does "#" refer to? It is referred to zero. We have added this information

in Table S2.

Please also note the supplement to this comment:
https://www.geosci-model-dev-discuss.net/gmd-2018-196/gmd-2018-196-AC1-supplement.pdf

**Supplement:**

General Comments

The manuscript by Yan et al. describes the implementation and impacts of an updated scheme for oxidation of aromatics (SAPRC-11) in the GEOS-Chem chemical transport model. The authors have provided a comprehensive overhaul to the previously very simplified benzene, toluene, and xylene chemistry. The updates are described in sufficient detail to allow reproducibility. The new simulation has been evaluated using both aircraft and surface observations and overall shows good agreement for aromatics and a reduction in model bias for ozone. The authors also quantify the impacts on related species including NOx, OH, and ozone, and show that there are small global impacts but significant regional impacts (especially over anthropogenic source regions). Overall, the updated chemistry is a valuable and important addition to a widely used global chemical transport model. The paper is generally well written, well structured, and easy to follow. The content and presentation are well suited to GMD, and I recommend publication once the following comments have been addressed.

We thank the reviewer for comments, which have been incorporated to improve the manuscript.

1. Model-observation comparisons should include the Base simulation

Section 4 compares the SAPRC (updated) simulation to the observations and discusses differences and biases. However, the Base (original) simulation is never compared to the aromatics observations. There is a brief comparison to the ozone observations, although this is buried in Section 5.3. To clearly show the impacts of the new chemistry on the simulation, both the Base and SAPRC simulations should be compared to the observations in Section 4. The Base simulation should be added to Figures 2-5, and the discussion currently on Page 12 lines 29-39 should be moved to Section 4 (along with Table 3).

Thanks for the comment from referee. We have added the modeled spatial distributions of annual mean surface (revised Figure 12) and zonal average latitude-altitude distributions of annual mean (revised Figure 13) benzene, toluene, and xylene simulated in the Base case for the year 2005. Also shown in these figures are the respective changes from Base to SAPRC. These two figures show that the changes from Base to SAPRC in annual average surface aromatics and zonal average aromatics are less than 2% for individual species. The differences between Base and SAPRC is much smaller than the modeled bias in SAPRC compared to aromatics observations. Thus we have kept the ozone comparison with Base and SAPRC in Sect.

5.3 to show the effects from SAPRC on ozone simulation.

2. SI tables should clearly identify new vs. updated species/reactions and should be consistent with GEOS-Chem nomenclature.

Table S1 states it provides a list of "new model species", but several of these are existing species in GEOS-Chem. This should be clarified, and could be done by changing "new" to "relevant" in the table caption and adding a column for "New or Existing" to the table.

In the revised Table S1, we have identified new vs. existing species by changing "new" to "relevant" in the table caption and adding a column for "New or Existing" to the table.

Similarly, Table S2 states it lists "new reactions and rate parameters", but again some reactions are currently in GEOS-Chem (presumably the rate parameters have been updated). Clarity is needed around what is new in the mechanism.

In the revised Table S2, we have identified new vs. updated reactions by changing "new" to "relevant" in the table caption and adding a column for "New or Updated" to the table. The updated reaction is meant to update the rate parameters.

Finally, the species names in the SI do not match the GEOS-Chem conventions for existing species. Just a few examples of what GEOS-Chem uses: MO2 (not MEO2), CH2O (not HCHO), ALD2 (not CCHO), and many more. This work will be much more usable by the GEOS-Chem community if the species list is updated to match. Existing species names are given at http://wiki.seas.harvard.edu/geos-chem/index.php/ Species_in_GEOS-Chem.

The SI tables should be updated where relevant to match.

We have updated the species list in Table S1 and Table S2 to match the GEOS-Chem conventions for existing species.

3. More details are needed to understand and be able to reproduce the model observation comparisons

Section 3 describes the observations used and, to some extent, the method in which the model was sampled for the comparisons. Some details are missing here that would be necessary for one to reproduce this work. Specifically, I had the following questions:

• CALNEX observations are at 1 second / 100 m resolution (pg 6, lines 20-21). This is

much higher resolution than the model (2x2.5 degrees, timestep on order of minutes). Have the aircraft data then been averaged to the model resolution? If not the statistics will be biased by comparing multiple observation points to a single model grid point, especially as there will not be equal observation points in a given gridbox / timestep.

We have added the information in the revised Sect. 3.1: "For comparison to the model results, we averaged the high temporal-spatial resolution observations to the model resolution."

• For CARIBIC comparisons model output is sampled along the flight track (pg 6, lines 31-33). If this is the case, then why are model annual means used for the comparison? Shouldn't these be treated in the same way as CALNEX?

We first averaged the measurements to the model output resolution. Then in comparison, we use annual means of observations and model data along the flight track. In the revised description, we have added the details: "To evaluate our results, measurements are averaged to the model output resolution. Vertically, results from GEOS-Chem model simulations at the 250 hPa level are used to compare with observations between 200–300 hPa. Then the annual means of observations and model data sampled along the flight tracks are used in the comparison."

• Why are monthly means used for EMEP comparisons but annual means used for EEA comparisons (pg 7, lines 11-12 and 21)?

We used monthly means for EMEP comparisons but annual means for EEA comparisons, mainly because that the EEA measurements have much more missing data than the EMEP observations.

• Why are urban and suburban sites excluded from EEA comparisons (pg 7 lines 18-19) but not excluded from AQS comparisons (pg 10, line 2)?

Based on the comment from referee#2, in the revised text, we have removed the model evaluation with AQS ozone measurements, because that it is inappropriate to directly compare AQS ozone observations near the surface ($\sim$ 10 m) to GEOS-Chem ozone at 65 m height with 2x2.5 deg horizontal resolution.

• For KCMP, the paper specifies use of hourly observations (pg 7, line 26); are hourly model values also used?

We averaged the hourly observations to monthly values and then compared to the monthly model results. We have added the information in the revised sentence: "We averaged the hourly observations of benzene, toluene and $C_8$ (xylenes + ethylbenzene; here consistent with the model speciation) aromatics to monthly values and then used

for our model evaluation."

• For KCMP sampling at 185m (pg 7, line 25), what box is the model sampled from, and how does that model layer compare to the 185m sampling height? I presume it wouldn't be the lowest model layer, since that is centred at 65m (pg 7 line 36).

We have added the information at the end of this paragraph: "Monthly mean simulations at the 990 hPa level (~190 m) are used for comparison."

• Why are so many more EMEP sites used for ozone (130) than for aromatics (8-14) (Table 1)?

It is because that aromatics downloaded from EMEP (http://www.nilu.no/projects/ccc/emepdata.html) are much less than ozone measurements.

• It would be useful to include a table providing sites and location information in the SI, especially since some stations have been excluded. This is probably not feasible for the large number of ozone sites, but would be for the aromatics data.

The sites and location information of aromatics data used here are described in detail in Cabrera-Perez et al., 2016 who download the raw data and provide the collated data. We have added this information in the revised text: "Further details of the sites and location information of EEA (and EMEP) used here are described in Cabrera-Perez et al., 2016."

4. Difference maps should be included for benzene, toluene, and xylene

Although they are the focus of the paper, no maps of the aromatics spatial distribution are shown (except emissions), although they are hinted at on pg 11, lines 25-27. It seems to me critical to include figures analogous to Fig. 6 and 7 but for benzene, toluene, and xylene.

We have added the modeled spatial distributions of annual mean surface (revised Figure 12) and zonal average latitude-altitude distributions of annual mean (revised Figure 13) benzene, toluene, and xylene simulated in the Base case for the year 2005. Also shown in these figures are the respective changes from Base to SAPRC.

5. Some comments are needed on the likely impact of changing aromatic emissions

Anthropogenic aromatics emissions are from the Year 2000, while simulations and observations are for 2005 and 2010-2011. There are likely to have been significant changes in emissions and their distributions over the decade (briefly noted by authors

on pg 8, lines 20-21). This is not a problem per se, but the paper would benefit from discussion of the likely changes and how they would benefit the results shown here (i.e. have aromatics gone up, in which case this work provides an upper limit? or the opposite?).

Thanks for the comment from referee. We have added discussion in the revised Sect. 4.1: "Anthropogenic aromatics emissions are reported to have significant changes in emissions and their distributions over the decade by EDGARv4.3.2 (Crippa et al., 2018; http://eccad.aeris-data.fr/#DatasetPlace:EDGARv4.3.2$DOI). It shows that the total aromatics emission from anthropogenic source are enhanced by 5% (2005) and 14% (2011) compared to the year 2000. The model bias would be partly benefit from this emission increase with enhanced modeled mixing ratios of benzene and toluene."

Specific Comments

Pg 2, lines 17-18: can some references be provided to back this up?

We have added the references of Lewis et al., 2013 and Cabrera-Perez et al., 2016.

Pg 2, lines 29-30: is the overestimate global, or region-specific?

We have revised this sentence: "However, some systematic biases can occur, most commonly an overestimation over the northern hemisphere"

Pg 2, line 35: the introduction has jumped from models in general to GEOS-Chem specifically, so GEOS-Chem needs some introduction here

We have revised this sentence: "Another motivation for the modeling comes from recent updates in halogen (bromine-chlorine) chemistry, which when implemented in GEOS-Chem, a global chemical transport model being used extensively for tropospheric chemistry and transport studies (Zhang and Wang, 2016; Yan et al., 2014; Shen et al., 2015; Lin et al., 2016), decrease the global burden of ozone significantly (by 14%; 2–10 ppb in the troposphere) (Schmidt et al., 2017)."

Pg 3, line 4: It would be better to use the updated GEOS-Chem versioning, which would make this version 12.0.0 rather than 11-02 (although technically the same).

We have modified the version.

Pg 3, line 9: I feel the equation would be easier to understand as aromatic + OH + NO = inert tracer (rather than "-NO" on the right-hand side)

We have modified the equation as aromatic + OH + NO = inert tracer

Pg 3, lines 31+: what model time step is used?

We have added the time step information: "The chemistry time step is 0.5 h, while the transport time step is 15 min in the model."

Pg 4, lines 10-17: unclear why CO emissions are discussed here when CO is not a focus of the paper (and never shown later). If included here, would need to also include non-anthro CO sources (e.g. chemical production). Also, emissions from ships and aircraft missing. It might make sense to start this section with the NMVOC emissions rather than CO/NOx as they are the focus.

We have removed the CO emission description and moved the NOx emission behind the NMVOC emission description.

Pg 4, line 28: please specify species for the "aromatics" source – is this just benzene + toluene + xylene, or are other species included? Also "(71 Tg C)" can be deleted as it is given in the previous sentence (line 27).

We have added the species for the "aromatics" source and removed the "(71 Tg C)" in this sentence: "On a carbon basis, the global aromatics (benzene + toluene + xylenes) source accounts for ~ 23% (16 TgC) of the total anthropogenic NMVOCs."

Pg 5, line 15: ". . . which is consistent with the recent literature." More details are needed. What specifically does the SAPRC-11 mechanism reconcile that is/was missing from other mechanisms?

We have added some details of major updates in the SAPRC-11: "SAPRC-11 is an updated version of the SAPRC-07 mechanism (Carter and Heo, 2013) to give better simulations of recent environmental chamber experiments. The SAPRC-07 mechanism underpredicted NO oxidation and $O_3$ formation rates observed in recent aromatic-$NO_x$ environmental chamber experiments. The new aromatics mechanism, designated SAPRC-11, is able to reproduce the ozone formation from aromatic oxidation that is observed in environmental chamber experiments (Carter and Heo, 2013)."

Pg 5, lines 21-33: I find this description hard to follow and hard to relate to what is in the tables in the SI. I think it would be helpful to give an example that traces the oxidation of one aromatic through these different production pathways.

We have modified the description by taking toluene as an example: "In SAPRC-11, taking toluene as an example in Table S2, the reactions following abstraction lead to three different formation products: an aromatic aldehyde (represented as the *BALD* species in the model), a ketone (*PROD2*), and an aldehyde (*RCHO*). The largest yield of toluene oxidation is the reaction after OH addition of aromatic rings. The

OH-aromatic adduct is reaction with $O_2$ either forming $HO_2$ and a phenolic compound (further consumed by reactions with OH and $NO_3$ radicals), or to form an OH-aromatic-$O_2$ adduct. The OH-aromatic-$O_2$ adduct further undergos two competing unimolecular reactions to ultimately form OH, $HO_2$, an α-dicarbonyl (such as glyoxal (*GLY*), methylglyoxal (*MGLY*) or biacetyl (*BACL*)), a monounsaturated dicarbonyl co-product (*AFG1*, *AFG2*, the photoreactive products) and a di-unsaturated dicarbonyl product (*AFG3*, the non-photoreactive products) (Calvert et al., 2002)."

Pg 6, lines 7-12: This is a little confusing and would suggest rephrasing. Is there a separate 6-month spin-up for each scenario (Base and SAPRC)? Is July-December 2009 also a spin-up period? For the sentence about initial conditions, does this mean that there is a 4◦x5◦ spin-up run from Jan-Jun 2004 followed by a 2◦x2.5◦ spin-up run for Jul-Dec 2004?

We have revised these sentences: "Both simulations (Base and SAPRC) at 2.5° long. × 2° lat. are conducted from July 2004 to December 2005, allowing for a 6-month spin-up for our focused analysis over the year of 2005 based on the available observations (Sect. 3). Initial conditions of chemicals are regridded from a simulation at 5° long. × 4° lat. started from 2004 with another spin-up run from January to June 2004. For comparison with aromatics observations over the US in 2010–2011 (Sect. 3), we extend the simulations from July 2009 to December 2011 with July-December 2009 as the spin-up period."

Pg 7, lines 13-14: The model speciation of xylenes should be clarified in the earlier section 2.2 about the mechanism.

We have added the model speciation of xylenes in revised Sect. 2.1: "Here the model speciation of xylenes includes m-xylene, p-xylene, o-xylene and ethylbenzene (Hu et al., 2015)."

Pg 8, lines 13-21: The model-observation difference would be a useful metric to include in Table 1.

We have added the calculation of model-observation difference in revised Table 1. It is the MRB (relative bias of model results) defined as: (MMOD – MOBS)/MOBS.

Pg 8, line 30: Why are these 6 sites the ones used?

We have added the reason: "The sites are chosen as the first six stations with largest amount of data."

Pg 9, lines 13, 19: what are the lifetimes for benzene and toluene?

The lifetime of benzene is between several hours and several days, and toluene is between several days and several weeks irrespective of the time of year (http://www.nzdl.org/gsdlmod?a=p&p=home&l=en&w=utf-8). We have added the lifetime for toluene in the revised Sect. 4.2.

Pg 9, lines 23-34: any comment about the different profile shapes in the lower troposphere? What about the overly rapid benzene drop-off with altitude? Does that imply the modelled benzene lifetime is too short?

Thanks for this comment from referee. We have added the discussion in the revised Sect. 4.2: "The different profile shapes in the lower troposphere for benzene, toluene and $C_8$ aromatics are mainly due to their different emissions and lifetime. The modeled overly rapid aromatics drop-off with altitude probably implies the modelled aromatics lifetime is short."

Pg 10, line 25: any comment on why winter shows an increase when the other seasons do not?

We have added the discussion in the revised Sect. 5.1: "This winter increase versus decline in other seasons is probably attributed to the weaken photochemical reactions involving $NO_x$ in winter."

Pg 10, line 28: "The free tropospheric increases are largest in the remote northern regions" – I don't see this in Fig. 7. Instead it looks like the NO increases are about the same from 30S-90N.

We have revised this sentence as: "The free tropospheric NO increases are about the same from 30S-90N".

Pg 10, lines 31-32: Rephrase this sentence as the start suggests it is about the surface NO2 but then it ends with "throughout the troposphere."

We have revised this sentence as: "Figure 6 shows that simulated surface $NO_2$ mixing ratios in the SAPRC scenario are enhanced over most locations across the globe".

Pg 11, lines 1-3: Because of the different color scales, the overall NOx changes are not obvious in Figs 6 and 7. I'd suggest adding another panel to show the total NOx change.

We have added the modeled spatial distributions of annual mean surface NOx (revised Figure 7) and zonal average latitude-altitude distributions of annual mean (revised Figure 8) NOx simulated in the Base case for the year 2005. Also shown in these figures are the respective changes from Base to SAPRC.

Pg 11, lines 4-7: any comments on what is driving the NO3 global decreases and regional increases?

We have added the discussion following this sentence: "The $NO_3$ global decreases are mainly due to the consumption of $NO_3$ by reaction with the aromatic oxidation products. However, the $NO_3$ regional increases are probably caused by the enhanced regional atmospheric oxidation capacity."

Pg 11, lines 8-11: Table 2 and the associated discussion in the text would be easier to follow if it were presented as a figure (e.g. a set of bar charts) rather than a densely packed table. Also, at the moment it includes species that are not discussed elsewhere in the text.

Thanks for this comment from referee. We have kept the table in the revised manuscript, mainly because that the amount of data in Table 2 is large to be difficult presented as a figure and be also difficult to show the specific value in the bar charts. In the revised Table 2, we have removed the calculation results of species ($H_2O_2$ and $N_2O_5$) that are not discussed in the text.

Pg 11, lines 31: Might be useful for this discussion to include the OH/HO2 ratio in the table (or figure)

We have included the OH/HO2 ratio in the revised Table 2.

Pg 12, lines 10-11: Please comment on why the ozone declines in biomass burning regions. Why have these changed in ways that are different from anthropogenic dominated regions? How can you tell that the changes are induced by biomass burning dominance rather than biogenic emissions dominance? If the former, I'm surprised not to see the same effects in boreal regions and in southern Africa.

Based on the recent data analysis, we cannot yet comment on why the ozone declines in regions dominated by biomass burning or biogenic emissions. We have revised this sentence to include the specific regions: "ozone declines in regions of South America, Central Africa, Australia and Indonesia over the tropics (30°S–30°N)." The reasons for the ozone decline are discussed below: "These decreases are probably related to the upward transport of aromatics by tropical convection processes. The aromatics transported to the upper troposphere may cause net consumption of tropospheric OH and $NO_x$, which can further reduce ozone production."

Pg 12, lines 13-22: The reasons for the ozone increases are described, but what is causing the ozone decreases?

The reasons for the ozone decline are discussed below: "These decreases are probably

related to the upward transport of aromatics by tropical convection processes. The aromatics transported to the upper troposphere may cause net consumption of tropospheric OH and NO$_x$, which can further reduce ozone production."

Pg 12, lines 27-28: Simulated production and loss rates could be used to test this.

Thanks for this comment from referee. Regretfully, we did not output the modeled results of production and loss rates.

Pg 12, lines 37-39: I think the conclusion here is that the halogen chemistry would bring the US ozone back down to the point that addition of aromatics would be a net improvement. If that's so, please make that point explicit. It also left me wondering what the impacts of the halogens would be outside of the US, where the biases shown in this work are already negative – would they become worse?

Based on the comment from referee#2, in the revised manuscript, we have removed the model evaluation with AQS ozone measurements and the discussion of halogen chemistry, because that it is inappropriate to directly compare urban and suburban AQS ozone observations near the surface ($\sim$ 10 m) to GEOS-Chem ozone at 65 m height with 2x2.5 deg horizontal resolution.

Pg 13, lines 6-9: It would be worth adding a panel to Figs. 8 and 9 to show the changes to the odd oxygen family. A panel for PAN would also be useful for the subsequent discussion.

We have added the modeled spatial distributions of annual mean surface PAN (revised Figure 7) and zonal average latitude-altitude distributions of annual mean PAN (revised Figure 8) simulated in the Base case for the year 2005. Also shown in these figures are the respective changes from Base to SAPRC. For the odd oxygen family (Ox), they are shown in revised Figure 10 and Figure 11.

Pg 13, line 25: which "organic nitrates" are referred to here? Is this PAN and analogues (PBZN)? Or does this refer to other organic nitrates like alkyl nitrates? It is not clear where in Table S2 one is meant to look for the chemistry of these nitrates.

We have added the specific species and the referred reactions shown in Table S2 in the revised sentence: "In addition, production of organic nitrates (*PBZN* (reactions of BR30 and BR31 in Table S2) and *RNO3* (PO36)) in the model with SAPRC aromatics chemistry".

Pg 13, lines 26-29: what NOx recycling is assumed in the model? Is this an effect that the authors have looked at (if so, can it be shown or described in more detail?), or does this refer to knowledge from existing literature (if so, references are needed. . .)?

We have changed the "recycling of NO$_x$" to "such re-release of NO$_x$" in the revised sentence. The re-release of NO$_x$ process have described in the former sentence: "In addition, production of organic nitrates (*PBZN* (reactions of BR30 and BR31 in Table S2) and *RNO3* (PO36)) in the model with SAPRC aromatics chemistry may also explain the increase in ambient NO$_x$ in the remote regions, due to the re-release of NO$_x$ from organic nitrates (as opposed to removal by deposition)."

Figure 2: Would be easier to interpret if common sites were aligned for the 3 species. (e.g. Zingst common between benzene and xylene, so move to upper left for xylene to match location for benzene, etc.)

We have moved the common sites to be aligned for the three species in the revised Figure 2.

Technical Comments

Title: GMD requires specifying model version number in addition to name ("GEOSChem version 9-02")

We have added the model version into the title.

Pg 5, line 5: suggest changing "true" to "the case"

Have changed.

Pg 5, line 6: change "v11-02" to "12.0.0" if changed above

Have changed.

Pg 5, line 10 (and elsewhere): the Carter and Heo (2013) reference is missing from the reference list

We have added the Carter and Heo (2013) reference in the reference list.

Pg 5, line 35: change "xylenols, phenols (XYNL)" to "xylenols and phenols (XYNL)" since XYNL represents both species.

Have changed.

Pg 6, line 8: suggest changing "based on the available observations" to "for comparison to the available observations"

Have changed.

Pg 6, line 18: suggest deleting "over the US" – this is too broad and already clear from the mention of California in the previous line.

Have deleted "over the US".

Pg 7, line 7: change "though" to "through"

Have changed.

Pg 7, line 8: suggest changing "boundaries" to "geographic boundaries" (to clarify that this is not flux through e.g. air-land boundaries)

Thanks for comment from referee. We have changed.

Pg 7, line 8: suggest changing to "locates measurement sites in locations where there are minimal. . ."

Have changed.

Pg 7, line 21: change "site" to "sites"

Have changed.

Pg 7, line 22: would be useful to add the location for the KCMP tall tower (e.g. US state?). Also does KCMP stand for something? Acronym is not defined.

We have added the location in the revised sentence: "The KCMP tall tower measurements (at 44.69°N, 93.07°W, Minnesota, US) have been widely used for studies". The the KCMP is the current Minnesota Public Radio.

Pg 8, line 2: suggest changing "part" to "section"

Have changed.

Pg 8, line 5: suggest changing "To do" to "For"

Have changed.

Pg 10, line 1: suggest deleting "relatively"

Have deleted.

Pg 13, line 33: suggest changing "give" to "provide"

Have changed.

Figure 7: caption error; missing reference to NO3 and to middle plots

We have added the reference to NO3 and to middle plots in caption of Figure 7.

Figures 6-9: are these annual means? Which model year?

Yes, they are annual means and for the year 2005. We have added the information in the captions.

Table S1: Benzene, Toluene, and Xylene missing from species list

We have added these three species in the revised Table S1.

Table S2: What does "#" refer to?

It is referred to zero. We have added this information in Table S2.

**Anonymous Referee #2**

This paper reported an excellent timely effort updating aromatic VOC chemistry in GEOS-Chem, a widely used global chemistry model. The effort is very useful for the community given the importance of aromatics in regional and global chemistry and the potential limitation of the existing chemical mechanism included in GEOS-Chem. The paper describes the motivation, methodology in a very clear fashion. The key model results (e.g., NOx, HOx, ozone) are selected appropriately and discussed thoroughly, and are interpreted carefully by recognizing both the strengths and the potential limitations of the model setup and input data. A very comprehensive model evaluation has been carried out using data from multiple global and regional networks/programs. I recommend publication after my following comments are considered.

We thank the reviewer for comments, which have been incorporated to improve the manuscript.

Major comments

- The use of AQS ozone data in model evaluation is inappropriate and should be removed

It is simply inappropriate to directly compare urban and suburban AQS ozone

observations near the surface (∼ 10 m) to GEOS-Chem ozone at 65 m height with 2x2.5 deg horizontal resolution. The model evaluation results using AQS data is not only meaningless but also misleading, especially when these results are discussed along with other networks in remote environments, where the model evaluation is actually appropriate and meaningful. Thus, I strongly suggest the authors remove the model evaluation with AQS ozone and focus on using networks over rural and clean environments.

Thanks for the comment from referee. In the revised manuscript, we have removed the model evaluation with AQS ozone measurements.

- The adoption of SAPRC-11 and uncertainties in knowledge of aromatic chemistry

The paper describes the SAPRC-11 mechanism itself in detail and the method to include it into GEOS-Chem clearly. However, it is yet to be more clear why it is chosen instead of other options, such as the condensed MCM mechanism. One thing about SAPRC is the use of maximum ozone formation as a primary metric in the chamber experiment benchmark, and the mechanism has been primarily used and evaluated in regional CTMs such as CMAQ and CAMx, at much finer resolution (i.e., a few kilometers). I think the present paper is the first to use it in a global model. Therefore, the authors should have some words justifying the approach. Also, are there other considerations behind the simplified GEOS-Chem aromatic chemistry, in addition to minimizing the number of reactions? Moreover, it should be noted that our knowledge about the very complex aromatic chemistry itself is not complete. For instance, how would the uncertainties in the yields of di-carbonyls and radical recycling affect the mechanism and the model simulations? The simplified chemistry in GEOS-Chem does not have radical cycling, but are there any assumptions/uncertainties in SAPRC-11 about radical cycling that might have impact on the results too?

Adding some discussions on these above questions would make the paper even stronger.

Thanks for the comment from referee. We have added discussion in the revised Sect. 5.4: "SAPRC is a highly efficient and compact chemical mechanism with the use of maximum ozone formation as a primary metric in the chamber experiment benchmark. The mechanism has been primarily used and evaluated in regional CTMs such as CMAQ and CAMx, at much finer resolution (i.e., a few kilometers). Our study has significant application to use it in a global model. Implementing SAPRC-11 aromatic chemistry would add ~3% more computational effort in terms of model simulation times.

SAPRC is based on lumped chemistry, which is partly optimized on empirical fitting

to smog chamber experiments that are representative to one-day photochemical smog episodes typical of, for example, Los Angeles and other US urban centers. However, SAPRC-11 gives better simulations of ozone formation in almost all conditions, except for higher (>100 ppb) $NO_x$ experiments where $O_3$ formation rates are consistently over predicted (Carter and Heo, 2013). This over prediction can be corrected if the aromatics mechanism is parameterized to include a new $NO_x$ dependence on photoreactive product yields, but that parameterization is not incorporated in SAPRC-11 because it is inconsistent with available laboratory data.

Other option, such as the condensed MCM mechanism, which are based upon more fundamental laboratory and theoretical data and used for policy and scientific modelling multi-day photochemical ozone formation, is experienced over Europe by Cabrera-Perez. (2016). Our results are consistent with the simulation of EMAC model implemented with a reduced version of the MCM aromatic chemistry. Moreover, aromatic chemistry is still far from being completely understood. For example, Bloss et al., (2005) show that for alkyl substituted mono-aromatics, when comparisons to chamber experiment over a range of VOC/$NO_x$ conditions, the chemistry under predicts the reactivity of the system but over predicts the amount of $O_3$ formation (model shows more NO to $NO_2$ conversion than on the experiments)."

Minor comments

P2, L19-L21: "Despite the potentially important influence of aromatic compounds on global atmospheric chemistry, their effect on tropospheric ozone formation in polluted urban areas remains largely unknown." "Unknown" is an overstatement of the issue to me. Aromatic VOCs have long been recognized as a key player in urban photochemistry, forming PAN and ozone, and SOA, despite the uncertainties with the chemistry (and emissions).

We have revised this sentence as: "Despite the potentially important influence of aromatic compounds on global atmospheric chemistry, their effect on tropospheric ozone formation in polluted urban areas is less analyzed with the model simulation."

P2, L21-L22: "The main source and sink processes of tropospheric ozone are photochemical production and loss, respectively (Yan et al., 2016)" Other references such as textbook by Seinfeld and Pandis (2006) would be more appropriate in this sentence.

We have added two more references of Seinfeld and Pandis (2006) and Monks et al. (2015) in the revised text.

P2, L33: ". . . including the parameterization of small-scale processes and their feedbacks to global-scale chemistry (Yan et al., 2014; Yan et al., 2016)." Other

references should be added in addition to these two.

We have added two more references of Chen et al. (2009) and Krol et al. (2005) in the revised text.

P5, L27: "The OH-aromatic adduct is reaction with O2. . ." This sentence needs rephrase.

We have revised this sentence as: "The OH-aromatic adduct is reaction with $O_2$ to form an OH-aromatic-$O_2$ adduct or $HO_2$ and a phenolic compound (further consumed by reactions with OH and $NO_3$ radicals)."

P6, L13: Have the authors considered evaluating species other than ozone and aromatics, such as aircraft measurements of HOx (CalNex probably has some HOx measurements)?

Thanks for the comment from referee. Regretfully, we have no measurements of HOx from CalNex.

P7, L32: Data download link does not work (last access 9/26/18) http://aqsdr1.epa.gov/aqsweb/aqstmp/airdata/download_files.html

We have removed the AQS ozone data analysis based on the first major comment above.

P7, L36: see my first major comment.

Thanks for the comment from referee. We have removed the model evaluation with AQS ozone measurements.

P12, L30: The discussions at AQS sites should be removed.

Have removed.

P13, Section 5.4: See my second major comment. I suggest adding discussions of uncertainty in knowledge of aromatic chemistry and the considerations and assumptions in SAPRC-11.

We have added discussion in the revised Sect. 5.4. Please see details in the response of major comment 2.

Table 2: I suggest add numbers for NH and SH

We have added in the revised Table 2.

**Anonymous Referee #3**

This paper describes the implementation of the (State-wide Air Pollution Research Center) SAPRC-11 representation of BTEX mono-aromatic chemistry into the 9-02 version of the GEOS-Chem global chemical transport model. This is timely, given the importance of aromatic chemistry in the global atmosphere, with respect to air quality (i.e. ozone and other secondary photochemical pollutants) and secondary organic aerosol formation. Model evaluations have been carried out against a significant, wide ranging observational database (both long term ground and aircraft flight path measurements) of aromatics and ozone concentrations. Model analysis of the effects of the new chemistry on the important model outputs of O3, NOx and HOx have been carried out and discussed with respect to global and regional biases.

Overall, this paper is reasonably well written (although lacking in some detail, especially with respect to the specific aromatic chemistry implemented – see discussion) and will be useful to the global CTM community. It is in good scope for GMD. I recommend publication after the following comments have been addressed.

We thank the reviewer for comments, which have been incorporated to improve the manuscript.

(1) More detailed description of aromatic photochemistry implemented (base case and updated aromatic chemistry).

It would be useful to the reader to have a more detailed description of the aromatic chemistry represented in the Base model as well as the SAPRC update. For example, a simplified schematic showing the structure of the different mono-aromatics and how reaction with OH leads to initial OH-adducts (and OH abstraction products from OH attack at the methyl groups) that can then convert to different ring retaining and ring opening products, though the representative RO2 species formed from subsequent reactions with O2 and NO, leading to significant O3 production. This chemistry is briefly discussed in the text, and in a way that is only understandable from an experienced GEOS-Chem user (form the base case at least) but should be given in more detail as this important chemistry is the subject of this paper.

Thanks for the comment from referee. We have described the aromatics chemistry of the base case in the introduction: "A simplified aromatic oxidation mechanism has previously been employed in GEOS-Chem (e.g., Fischer et al., 2014; Hu et al., 2015), which is still used in the latest version v12.0.0. In that simplified treatment, oxidation

of benzene (B), toluene (T), and xylene (X) by OH (Atkinson et al., 2000) is assumed to produce first-generation oxidation products (xRO$_2$, x = B, T, or X). And these products further react with hydrogen peroxide (HO$_2$) or nitric oxide (NO) to produce LxRO$_2$y (y = H or N), passive tracers which are excluded from tropospheric chemistry. Thus in the presence of NO$_x$, the overall reaction is aromatic + OH + NO = inert tracer. While such a simplified treatment can suffice for budget analyses of the aromatic species themselves, it does not capture ozone production from aromatic oxidation products."

In the revised text, we have taken toluene as an example to describe the SAPRC-11 aromatics chemistry: "As discussed by Carter (2010a, b), aromatic oxidation has two possible OH reaction pathways: OH radical addition and H-atom abstraction (Atkinson, 2000). In SAPRC-11, taking toluene as an example in Table S2, the reactions following abstraction lead to three different formation products: an aromatic aldehyde (represented as the *BALD* species in the model), a ketone (*PROD2*), and an aldehyde (*RCHO*). The largest yield of toluene oxidation is the reaction after OH addition of aromatic rings. The OH-aromatic adduct is reaction with O$_2$ to form an OH-aromatic-O$_2$ adduct or HO$_2$ and a phenolic compound (further consumed by reactions with OH and NO$_3$ radicals). The OH-aromatic-O$_2$ adduct further undergos two competing unimolecular reactions to ultimately form OH, HO$_2$, an α-dicarbonyl (such as glyoxal (*GLY*), methylglyoxal (*MGLY*) or biacetyl (*BACL*)), a monounsaturated dicarbonyl co-product (*AFG1*, *AFG2*, the photoreactive products) and a di-unsaturated dicarbonyl product (*AFG3*, the non-photoreactive products) (Calvert et al., 2002).

Formed from the phenolic products, the SAPRC-11 mechanism includes species of cresols (*CRES*), phenol (*PHEN*), xylenols and alkyl phenols (*XYNL*), and catechols (*CATL*). Due to their different SOA and ozone formation potentials (Carter et al, 2012), these phenolic species are represented separately. Relatively high yields of catechol (*CATL*) have been observed in the reactions of OH radicals with phenolic compounds. Furthermore, their subsequent reactions are believed to be important for SOA and ozone formation (Carter et al, 2012)."

Also, when discussing the SAPRC aromatic-ozone chemistry in Section 5.4, it would be useful to provide the basic photochemical ozone formation chemistry equations (including PAN formation) so that the discussion in the text can be followed more closely.

In the revised text, we have referenced the basic chemistry equations: "From Base to SAPRC, modeled PAN has been enhanced in a global scale (Fig. 8 and 9) via reactions of aromatic-OH oxidation products with NO$_2$ (equation of BR13 in Table S2). In the SAPRC-11 aromatics chemical scheme the immediate precursor of PAN (peroxyacetyl radical) has five dominant photochemical precursors. They are acetone

(CH3COCH3, model species: *ACET*), methacrolein (*MACR*), biacetyl (*BACL*), methyl glyoxal (*MGLY*) and other ketones (e.g., *PROD2*, *AFG1*). These compounds explain the increased rate of PAN formation. For example, the SAPRC simulation has increased the concentration of *MGLY* by a factor of 2. In addition, production of organic nitrates (*PBZN* (reactions of BR30 and BR31 in Table S2) and *RNO3* (PO36)) in the model with SAPRC aromatics chemistry may also explain the increase in ambient $NO_x$ in the remote regions, due to the re-release of $NO_x$ from organic nitrates (as opposed to removal by deposition). Due to such re-release of $NO_x$ from PAN-like compounds and also transport of $NO_x$, $NO_x$ increases by up to 5% at the surface in most remote regions and by ~1% in the troposphere as a whole. This then leads to increased ozone due to the effectiveness of ozone formation in the free troposphere."

(2) Discussion of uncertainties in the aromatic chemistry and comparisons with other, more detailed mechanisms.

There is little discussion about the development of the SAPRC chemical mechanisms, the uncertainties in the specific aromatic chemistry implemented and how the chemistry compares to other widely used detailed chemical schemes.

SAPRC was originally developed in order to model one day photochemical smog episodes typical of, for example, Los Angeles and other North American urban centres. SAPRC is a highly efficient and compact chemical mechanism, therefore can be implementation into CTMs, but is based on lumped chemistry, which is partly optimised on empirical fitting to smog chamber experiments that are representative to US one day conditions. Therefore, some discussion should be made with respect to applications of this optimised chemistry

outside these optimisation conditions – e.g. SH tropics. How does the SAPRC chemistry compare to more detailed chemical mechanisms, which are based upon more fundamental laboratory and theoretical data, which are used for policy and scientific modelling multi-day photochemical ozone formation that is experienced over Europe – e.g. the Master Chemical Mechanism?

It is also clear from the literature and atmospheric chamber model-mechanism comparisons that aromatic chemistry is still far from being completely understood. For example, Bloss et al., (2005) show that for alkyl substituted mono-aromatics, comparisons to chamber experiment over a range of VOC/NOx conditions that the chemistry under predicts the reactivity of the system but over predicts the amount of O3 produced (model shows more NO to NO2 conversion than on the experiments). How does the uncertainties in the fundamental aromatic chemistry effect the modelling shown here?

Thanks for the comment from referee. We have added discussion in the revised Sect.

5.4: "SAPRC is a highly efficient and compact chemical mechanism with the use of maximum ozone formation as a primary metric in the chamber experiment benchmark. The mechanism has been primarily used and evaluated in regional CTMs such as CMAQ and CAMx, at much finer resolution (i.e., a few kilometers). Our study has significant application to use it in a global model. Implementing SAPRC-11 aromatic chemistry would add ~3% more computational effort in terms of model simulation times.

SAPRC is based on lumped chemistry, which is partly optimized on empirical fitting to smog chamber experiments that are representative to one-day photochemical smog episodes typical of, for example, Los Angeles and other US urban centers. However, SAPRC-11 gives better simulations of ozone formation in almost all conditions, except for higher (>100 ppb) $NO_x$ experiments where $O_3$ formation rates are consistently over predicted (Carter and Heo, 2013). This over prediction can be corrected if the aromatics mechanism is parameterized to include a new $NO_x$ dependence on photoreactive product yields, but that parameterization is not incorporated in SAPRC-11 because it is inconsistent with available laboratory data.

Other option, such as the condensed MCM mechanism, which are based upon more fundamental laboratory and theoretical data and used for policy and scientific modelling multi-day photochemical ozone formation, is experienced over Europe by Cabrera-Perez. (2016). Our results are consistent with the simulation of EMAC model implemented with a reduced version of the MCM aromatic chemistry. Moreover, aromatic chemistry is still far from being completely understood. For example, Bloss et al., (2005) show that for alkyl substituted mono-aromatics, when comparisons to chamber experiment over a range of VOC/$NO_x$ conditions, the chemistry under predicts the reactivity of the system but over predicts the amount of $O_3$ formation (model shows more NO to $NO_2$ conversion than on the experiments)."

(3) Specific Comments

References are not in alphabetical order

We have reordered the references in alphabetical order.

How much more computational effort does implementing SAPRC-11 chemistry add in terms of model simulation times?

In the revised Sect. 5.4, we have added the information as: "Implementing SAPRC-11 aromatic chemistry would add ~3% more computational effort in terms of model simulation times."

Introduction – better referencing of the aromatic literature needed, e.g. Atkinson and Arey (2003) and Calvert et al., (2002).

We have added this two references into the revised introduction.

"Despite the potentially important influence of aromatic compounds on global atmospheric chemistry, their effect on tropospheric ozone formation in polluted urban areas remains largely unknown". This statement is simply not true. There is a large amount of literature on this subject and original policy based emission reactivity indexes such as MIR (which is based on SAPRC) and POCP (which is based on MCM) show the importance of aromatic chemistry to ozone formation in the US and Europe respectively.

We have revised this sentence as: "Despite the potentially important influence of aromatic compounds on global atmospheric chemistry, their effect on global tropospheric ozone formation in polluted urban areas is less analyzed with the model simulation."

"Current global CTMs reproduce much of the observed regional and seasonal variability in tropospheric ozone concentrations." This is a broad statement and needs to be qualified. Surely the very reason that you are carrying out this study is that this is not true?!

We have added further statement of model bias on ozone: "However, some systematic biases can occur, most commonly an overestimation over the northern hemisphere (Fiore et al., 2009; Reidmiller et al., 2009; Yan et al., 2016, 2018a, b; Ni et al., 2018)"

"GEOS-Chem" needs to be defined in more detail. References to v9-02 and v11-02 need to be added.

We have added more information of GEOS-Chem v9-02 in revised Sect. 2: "GEOS-Chem is a global 3-D chemical transport model for a wide range of atmospheric composition problems. It is driven by meteorological data provided from the Goddard Earth Observing System (GEOS) of the NASA Global Modeling Assimilation Office (GMAO). A detailed description of the GEOS-Chem model is available at http://acmg.seas.harvard.edu/geos/geos_chem_narrative.html." We have changed the recent version of v11-02 to v12.0.0 based on the comment from referee#1.

"SAPRC-11" also needs better defining

We have revised the introduction of SAPRC-11 in Sect. 2.2: "This work uses a more detailed and comprehensive aromatics oxidation mechanism: the State-wide Air Pollution Research Center version 11 (SAPRC-11) aromatics chemical mechanism. SAPRC-11 is an updated version of the SAPRC-07 mechanism (Carter and Heo, 2013) to give better simulations of recent environmental chamber experiments."

2.2. Updated aromatic chemistry – "Moreover,SAPRC-11 is able to reproduce the ozone formation from aromatic oxidation that is observed in environmental chamber experiments". Under what conditions? (VOC/NOx)

We have added this information in revised Sect. 2.2: "The new aromatics mechanism, designated SAPRC-11, is able to reproduce the ozone formation from aromatic oxidation that is observed in almost all environmental chamber experiments, except for higher (>100 ppb) $NO_x$ (Carter and Heo, 2013)."

3.2 Aromatic Surface Measurements – where is the KCMP tower? Define.

We have added the location of KCMP tower: "The KCMP tall tower measurements (at 44.69°N, 93.07°W, Minnesota, US) have been widely used for studies".

5.1 NOy Species – "Combing the changes in NO..." ???

"Combing the changes in NO..." is to discuss the NOx (NO + NO2) changes here; following paragraphs discuss the other NOy species.

5.2 OH and HO2 – "Compared to the Base simulation, OH increases slightly by 1.1% at the surface in the SAPRC simulation (Fig. 8 and Table 2)." Discussion of the observed deceases?

We have added description of deceases in the revised sentence: "Compared to the Base simulation, OH increases slightly by 1.1% at the surface in the SAPRC simulation, with that declines over the tropics (30°S–30°N) are compensated by enhancements over other regions (Fig. 8 and Table 2)."

"In these locations, the peroxy radicals formed by aromatic oxidation react with NO2 and HO2" – surely NO and HO2?

Have changed NO2 to NO.

"This in turn influences OH, as the largest photochemical sources of OH are the photolysis of O3 as well as the reaction of NO with HO2" – largest photochemical sources of OH in the model.

We have revised this sentence as: "This in turn influences OH, as the largest photochemical sources of OH in the model are the photolysis of O3 as well as the reaction of NO with HO2"

"Seasonally, a few surface locations see OH concentration increases of more than 10% during April−August (not shown), including parts of the eastern US, central Europe,

eastern Asia and Japan." There seem to be a few points in the text where interesting model results are eluded to but "not shown". Could some of these not be included in the supplementary?

We have added a figure in the revised supplementary to show the modeled spatial distributions of surface OH during April−August simulated in the Base case for the year 2005. Also shown is the respective relative changes (%) from Base to SAPRC.

5.3 Ozone – "The aromatics transported to the upper troposphere may cause net consumption of tropospheric OH and NOx, which can further reduce ozone production". How?

By reactions of aromatics with OH and NOx.

Could other atmospherically important species that are in aromatic chemistry be compared to the observations – specifically the detailed data sets from CALNEX – e.g. HOx, HCHO, PAN, Glyoxal and Methyl Glyoxal? These are all important tracers of active photochemistry.

Thanks for the comment from referee. Regretfully, we have no measurements of species other than aromatics (Benzene, Toluene and C8 aromatics) from CalNex.

**Comment from Executive editor**

Dear authors,

In my role as Executive editor of GMD, I would like to bring to your attention our Editorial version 1.1:

http://www.geosci-model-dev.net/8/3487/2015/gmd-8-3487-2015.html

This highlights some requirements of papers published in GMD, which is also available on the GMD website in the 'Manuscript Types' section:

http://www.geoscientific-model-development.net/submission/manuscript_types.html

We thank the Executive editor for comments, which have been incorporated to improve the manuscript.

In particular, please note that for your paper, the following requirements have not been met in the Discussions paper:

"The main paper must give the model name and version number (or other unique identifier) in the title."

"All papers must include a section, at the end of the paper, entitled 'Code availability'. Here, either instructions for obtaining the code, or the reasons why the code is not available should be clearly stated. It is preferred for the code to be uploaded as a supplement or to be made available at a data repository with an associated DOI (digital object identifier) for the exact model version described in the paper. Alternatively, for established models, there may be an existing means of accessing the code through a particular system. In this case, there must exist a means of permanently accessing the precise model version described in the paper. In some cases, authors may prefer to put models on their own website, or to act as a point of contact for obtaining the code. Given the impermanence of websites and email addresses, this is not encouraged, and authors should con- sider improving the availability with a more permanent arrangement. After the paper is accepted the model archive should be updated to include a link to the GMD paper."

Please include the version number of GEOS-Chem in the title of the revised manuscript. Additionally, please include information how to optain the GEOS-Chem Code into the Code Availability Section. Note, that it is not sufficient to only state that the code is available from author without stating reasons, why publication is not possible.

Yours,
Astrid Kerkweg

We have added the model version number in the revised title: "Global tropospheric effects of aromatic chemistry with the SAPRC-11 mechanism implemented in GEOS-Chem version 9-02"

We have added the code availability: "The GEOS-Chem code of version 9-02 used to generate this paper and the model results are available upon request. We are submitting the code for inclusion into the standard model. The revised aromatics chemistry will be incorporated in the current version 12.0.0 and the later versions."